# An optimization-based approach to track the Asian summer monsoon anticyclone across daily and interannual variability

Oleh Kachula<sup>1</sup>, Bärbel Vogel<sup>1</sup>, Gebhard Günther<sup>1</sup>, and Rolf Müller<sup>1</sup>

<sup>1</sup>Institute of Climate and Energy Systems (ICE-4), Forschungszentrum Jülich, Jülich, Germany

**Correspondence:** Oleh Kachula (o.kachula@fz-juelich.de)

#### Abstract.

The definition of the boundary of the Asian summer monsoon anticyclone (ASMA) in the upper troposphere-lower stratosphere (UTLS) (350 K-410 K) is a known challenge that highly impacts the information about the anticyclone's behaviour and affects the results when studying of its interannual variability. We present a novel method based on the absolute vortex moments that defines the ASMA boundary by solving an optimization problem. Here, we address the ASMA's climatology (1980-2023), interannual variability, the variability of the start and end dates and the duration of the anticyclone peak phase calculated with help of the defined novel method by using the ERA5 reanalysis provided by ECMWF. In addition, three individual years – 2017, 2022 and 2023 are highlighted during which aircraft campaigns took place to measure air inside the ASMA or its outflow (StratoClim, ACCLIP, PHILEAS). The interannual analysis is based on the anticyclone's centroid latitude and longitude, excess kurtosis, angle and aspect ratio using four isentropic surfaces: 350, 370, 390 and 410 K. Our findings show that the ASMA area decreases at 370, 390 and 410 K over the period 1980-2023 in contrast to previous studies. Further, we provide evidence of possible bimodality of the ASMA where the spatial distribution shows clustering of high-magnitude values (the Montgomery streamfunction values minus an optimized background value) around two centers on climatological data over 44 years as well as counting the number of days when two anticyclones (or two modes) where found simultaneously.

#### 1 Introduction

15

The Asian summer monsoon anticyclone (ASMA) is an important upper troposphere-lower stratosphere (UTLS) meteorological circulation pattern in the Northern Hemisphere during boreal summer. The anticyclone spans from East Africa to the western Pacific connected with strong convection occurring over South and East Asia (Randel and Park 2006; Park et al. 2008). The Asian monsoon system plays a major role in uplifting near-surface emissions e.g., anthropogenic pollutants or greenhouse gases to the UTLS (Lau and Kim 2022).

Within the ASMA trace gases as well as aerosols particles and their gas-phase precursors are confined and can be exported to the northern extra-tropical UTLS as well as to the global stratosphere (e.g., Vogel et al. 2016; Ploeger et al. 2017; Yu et al. 2017; Adcock et al. 2021; Lauther et al. 2022; Yan et al. 2019; Ungermann et al. 2016).

Understanding the inter-annual (and intra-seasonal) variability of the ASMA in terms of area, position, bimodality, shape and duration is therefore important for estimating whether climate change has altered this chemical transport pathway in the last decades (e.g., Lal et al. 2001; Kripalani et al. 2007; Turner and Annamalai 2012; Fadnavis et al. 2019; Basha et al. 2020).

25

Possible reasons for the variability of the Asian summer monsoon anticyclone are currently being discussed and include multiple large-scale climate phenomena, such as El Niño-Southern Oscillation (ENSO), quasi-biennial oscillation (QBO), Indian Ocean Dipole (IOD). ENSO in particular, a key driver of global climate variability, affects the ASMA strength (in terms of the residual Montgomery streamfunction values) and position (Basha et al. 2020; Kumar and Ratnam 2021).

Manney et al. (2021) show that in the last decades (1979-2018) the ASMA underwent noticeable interannual change: its area is growing, the anticyclone is shifted to the north, its formation starts earlier and the brake-up phase ends later.

To be able to study the dynamical properties of the Asian summer monsoon anticyclone it is necessary to formally define it (Manney et al. 2021). Here, similar to Manney et al. (2021), we characterise the ASMA by considering the Montgomery streamfunction (or potential)  $\mu$  on an isentropic surface. There is a debate how to define the edge of the anticyclone best (e.g., Ploeger et al. 2015). The definition impacts the moment quantities in the following analysis and the conclusions that could be drawn on its interannual variability. Previously the methodologies to define the ASMA were based on using several quantities, such as potential vorticity (Ploeger et al. 2015), although it provides only an enclosed contour in a very narrow range of potential temperatures, using gradient or anomaly fields of geopotential height (e.g. Zarrin et al. 2010; Barret et al. 2016; Nützel et al. 2016), and Montgomery streamfunction on isentropic surface (e.g. Popovic and Plumb 2001; Santee et al. 2017; Manney et al. 2021).

An additional difficulty for the definition of the edge of the ASMA is a "leaky" transport barrier which allows outflow out of the ASMA during the monsoon season (Dethof et al. 1999; Popovic and Plumb 2001; Garny and Randel 2013; Ploeger et al. 2013; Vogel et al. 2016). This outflow can be both westward and eastward and is sometimes referred to as eddy shedding (Popovic and Plumb 2001; Vogel et al. 2015; Riese et al. 2025). Further, the ASMA can potentially turn into a bimodal state (Zhang et al. 2002) introducing challenges to define the boundary.

Santee et al. (2017) use Montgomery streamfunction values to approximate the boundary of the ASMA at the position where the strong wind speed gradients appear; they select a value for each potential temperature. After calculating MSF background values (the terminology used in Matthewman et al. (2009) to describe the threshold), Manney et al. (2021) applied moments analysis to calculate the centroid latitude and longitude, aspect ratio and excess kurtosis.

Santee et al. (2017) and Manney et al. (2021) use a constant climatological MSF value for all years considered and throughout the entire monsoon season per isentropic surface. The main challenge is to describe the ASMA during development and break-up phases, where MSF background values might be too small to highlight the anticyclone or too large to provide the boundaries at all e.g., residual MSF values inside of the ASMA box might be zero or near zero. To handle these special cases, Manney et al. (2021) filters out data when the ASMA area is less than 1 % of the area of the hemisphere.

We propose a method to identify MSF background values for an individual point in time per isentropic surface in contrast to choosing one single value for one monsoon period. Thus in the approach presented here, an optimized MSF value to describe the ASMA boundaries will be presented reflecting the day-to-day variability of the ASMA. Our dynamical approach avoids some

subjectivity of selecting the most suitable MSF value to enclose the ASMA boundaries. Another advantage of the proposed method is that it can be used on any time scale, choosing individual days or specific hours.

After retrieving a set of optimized MSF background values that are used to enclose the ASMA boundaries, we apply a moments analysis similar to Manney et al. (2021). To be able to compare our results to those by Manney et al. (2021), we show climatological values from 1980 to 2023, but instead of taking a fixed length period within each year (such as JJA), we define the peak phase of the ASMA.

65

80

In 2017, 2022 and 2023 aircraft measurement campaigns: StratoClim, ACCLIP and PHILEAS respectively were conducted in the region of the AMA, therefore in addition to the climatology we provide the moments analysis for the years 2017, 2022 and 2023 separately to compare these selected years with the 44-years climatology.

Part of StratoClim was an aircraft measurements campaign conducted on the Indian subcontinent over the southern side of the Himalayas between 27 July and 10 August (e.g., Singer et al. 2022; Vogel et al. 2023; Stroh and StratoClim-Team 2025). During StratoClim a variety of trace gases and aerosol characteristics were measured for the first time up to 20 km altitude in the ASMA to characterise major processes which dominate particle and trace gas transport from surface sources into the lower stratosphere.

The Asian Summer Monsoon Chemical and Climate Impacts Project (ACCLIP) was a comprehensive airborne field campaign operated during the monsoon season in August 2022. The campaign operated two research aircraft and ground-based balloon sounding equipped with instruments to measure trace gases and aerosol content over the Western Pacific region. The scientific objectives of ACCLIP campaign were to investigate how the anticyclone affects transport pathways of uplifted air to the UTLS. The measurements provide the insights into the chemical content composition, ozone chemistry, aerosol radiative effects and the role of distribution of Asian pollutants (Pan et al. 2024; Pan et al. 2025; Smith et al. 2025).

PHILEAS airborne campaign was conducted between August and October 2023 using the German research aircraft HALO (Riese et al. 2025). The campaign collected data that give insight into how the Asian summer monsoon anticyclone transport pollutants into the Northern extra-tropics. The campaign was split into two phases over different regions: in early to mid-August during the first measurement phase, monsoon air containing pollutants over the Eastern Mediterranean, Israel and Jordan were investigated; in the second phase, during flights from Anchorage, Alaska, the transport of polluted air over the Pacific, Alaska, and Canada was probed. The measurements provide an opportunity to study the polluted monsoon air at higher latitudes and altitudes up to the extratropical lower stratosphere, gathering the data on e.g., water vapor, methane, ozone concentration as well as aerosol and its gas-phase precursors.

In section 2, we explain the methodology to determine the boundaries of the ASMA, describe the peak phase of the ASMA and underscore challenges of the method, its advantages and disadvantages. Also we show the comparison of the calculated MSF values with our novel method to Santee et al. (2017). Section 3 presents the horizontal distribution of the MSF residuals, climatological and individual time-series of the moments analysis, inter-annual variability of the ASMA area, centroid latitude and longitude.

#### 2 Data description and methodology

#### 2.1 The ERA5 dataset

For our study we use the ERA5 reanalysis (Hersbach et al. 2020) from the European Centre of Medium-Range Weather Forecasts (ECMWF) to calculate the Montgomery streamfunction. The range for interannual analysis covers the period 1980 to 2023, spanning April-October months within each year. We use a version of ERA5 with a resolution of 1° × 1°, referred to as ERA5 1° × 1° (similar to Ploeger et al. 2021; Konopka et al. 2022; Clemens et al. 2024; Vogel et al. 2024). ERA5 1° × 1° data are directly provided by the ECMWF on a 1° × 1° horizontal grid after down-scaling the original data provided on a 0.25° × 0.25° horizontal grid. The vertical resolution (137 vertical levels up to 0.01 hPa) is not changed and is the same as in the original ERA5 reanalysis. For our analysis we use ECMWF data at noon (12:00 UTC) but more frequent time-steps (say every 6 hours) are also possible. By using single points in time we tried to avoid any averaging routines because we wanted to provide a useful tool to study the ASMA during short time-scale campaigns.

#### 2.2 Methodology

In this work we use Montgomery streamfunction which is defined as

$$\mu = c_p T + \Phi \tag{1}$$

where,  $c_p$  is the specific heat capacity at constant pressure, T is temperature and  $\Phi = gz$  is the geopotential, with gravitational acceleration g and height z. We use a regular grid  $1^{\circ} \times 1^{\circ}$  resolution. To account for the variation in grid-cell area with latitude, we applied area weighting derived from Lambert Cylindrical Equal-Area projection in all our calculations related to both the proposed method and the ASMA spatio-temporal characteristics.

The analysis of the interannual and intra-seasonal variability of the Asian summer monsoon anticyclone depends on the choice of enclosing boundaries of the ASMA. The method proposed here to define the boundaries of the ASMA is based on the absolute vortex moment method that was already used to define the polar vortex Matthewman et al. (2009).

The absolute vortex moment  $M_{kl}$  is given by

115 
$$M_{kl}(\hat{\mu}, \mu_b) = \sum_{i=0}^{n-1} \sum_{j=0}^{m-1} (\hat{\mu}_{i,j} - \mu_b) y_i^k x_j^l \Delta y \Delta x,$$
 (2)

with

110

$$\hat{\mu}_{i,j} := \begin{cases} \mu_{i,j} & \text{where } \mu_{i,j} \ge \mu_b \\ \mu_b & \text{where } \mu_{i,j} < \mu_b \end{cases}, \tag{3}$$

where  $\mu_{i,j}$  is the Montgomery streamfunction (MSF) defined on a regular grid of the size  $n \times m$ , for i = 0, ..., n-1, j = 0, ..., m-1;  $\mu_b$  is a background value of MSF,  $y_i$  and  $x_j$  are latitude and longitude coordinates respectively,  $\Delta y$  and  $\Delta x$  are

stepsizes over latitude and longitude axes respectively; k and l are non-negative integers that can be 0, 1 or 2 depending on what we want to calculate. For example choosing k = l = 0 (e.g.,  $M_{00}$ ) gives us a sum over residual values of the MSF:

$$M_{00} = \sum_{i=0}^{n-1} \sum_{j=0}^{m-1} (\hat{\mu}_{i,j} - \mu_b) \Delta y \Delta x.$$

Because we use ERA5 reanalysis with  $1^{\circ} \times 1^{\circ}$  grid resolution  $\Delta x = 1$ ,  $\Delta y = 1$ , n = 181, m = 360, with

$$x_j := \begin{cases} j, & \text{if } 0 \le j \le 180\\ 180 - j, & \text{if } 181 \le j \le 359 \end{cases}$$

125 and

135

145

$$y_i := 90 - i$$
, if  $0 < i < 180$ .

The method proposed by Matthewman et al. (2009) (who also introduced the term "background value" for  $\mu_b$ ) allows the aspect ratio (the ratio between the longitudinal and latitudinal axes of the equivalent ellipse), the angle between the equatorial axis and the major axis of the ellipse (the calculation of which depends not only on the boundary of the ASMA but also on the residual MSF values. See Eq. A2 and A3 in the Appendix A), the excess kurtosis and the coordinates of the center of the ASMA (ellipsoid center coordinates) to be retrieved (see Appendix A). The excess kurtosis (EK) serves as a measure of how far the shape is from the ellipse and can be used to investigate the splitting events of the ASMA into two anticyclones or strong shedding events (e.g., Popovic and Plumb 2001; Vogel et al. 2015; Riese et al. 2025). The anticyclone area is calculated separately because it is not based on the moments definitions. The information about the method is given below. To calculate the moments quantities the method requires the background value  $\mu_b$ . Matthewman et al. (2009) use the mean PV value poleward 45° N as the background value to encircle the polar vortex.

In case of the ASMA it is not possible to choose a specific region that should be averaged to obtain the background value for MSF, because several large scale monsoon systems (e.g., over America and Africa) exist in addition to the ASMA in contrast to the case of the polar vortex. Averaging different regions and hence varying the background value leads to a change of the boundaries of the ASMA. Further, the ASMA is very variable in its shape and sometimes bimodal (Tibetan and Iranian mode) or trimodal (e.g. eddy shedding events) distributions occur (Zhang et al. 2002; Vogel et al. 2015; Nützel et al. 2016; Siu and Bowman 2020; Wang et al. 2022; Pan et al. 2024). This might cause issues with capturing the anticyclone and preserve unwanted noise. Thus the method of Matthewman et al. (2009) if not directly applicable to the ASMA.

The novelty compared to Manney et al. (2021) of the method proposed here consists of splitting the globe into the ASMA box and the rest, defining the ratio of the absolute vortex moments (when k and l=0) as objective function. Doing this we reformulate the task of enclosing the ASMA boundary as an optimization problem where we can investigate the objective function to determine an optimized background value  $\tilde{\mu}_b$ .

The ASMA box is defined based on our empirical knowledge that the anticyclone can be found in [0° N-90° N, 0° E-180° E] region and serves to separate the Asian monsoon from other monsoon systems occurring during boreal summer such as the

American or African monsoon. By introducing the ASMA box it allows us to calculate  $M_{00}^{\text{in}}$  and  $M_{00}^{\text{out}}$  using Eq. (2) (k = 0, l = 0). The idea is that the values of Montgomery streamfunction inside of the dominant anticyclone region are higher than majority of  $\hat{\mu}_{i,j}$  values outside of the ASMA box, so we can find such  $\mu_b$  that keeps  $M_{00}^{\text{out}}$  minimal while maximize  $M_{00}^{\text{in}}$ . Using this idea we introduce the objective function as ratio of absolute vortex moments inside and outside of the ASMA box. The objective function

155 
$$f(\mu_b \mid \hat{\mu}_{i,j}) = \frac{M_{00}^{\text{in}}(\hat{\mu}_{i,j}, \mu_b)}{M_{00}^{\text{out}}(\hat{\mu}_{i,j}, \mu_b) + 1},$$
 (4)

should be maximized to determine the optimized boundary. By adding "+1" (m<sup>2</sup> s<sup>-2</sup>) we avoid singularity because  $M_{00}^{\text{out}}$  can be zero by definition (Eq. (3)). The background value, that we are looking for, is given by argmax  $f(\mu_b \mid \hat{\mu}_{i,j})$ . Most optimization algorithms are designed to minimize the objective function, so we need to transform the problem to minimize  $-f(\mu_b \mid \hat{\mu}_{i,j})$ . The global minimum can be found then by using optimization algorithms such as Dual Annealing (Xiang et al. (2013)) or by explicitly sampling the objective function and determining its global minimum. The  $\hat{\mu}_{i,j}$  is a parameter in this case and during minimization it is fixed for one specific point in time (i.e., there is no time dependence in Eq. 4).

Minimizing the objective function

$$\underset{\mu_b \in \mathbb{R}^+}{\text{minimize}} \quad -f(\mu_b \mid \hat{\mu}), \tag{5}$$

allows us to determine the optimized background value  $\tilde{\mu}_b$  which we can use in Eq. (2) to calculate time-series of the moments quantities. Typical form of the objective function is shown in Figs. 1 and 2 for two particular days: 1 August 2008 and 6 September 2008 respectively.

We discuss two cases of MSF grid data:

175

- 1. Many of the grid values inside of the ASMA box are larger than values outside of the ASMA box.
- 2. Regions outside of the ASMA box have MSF values that are comparable or larger to the values inside of the ASMA box.
- Let's consider two specific days that correspond to the above described scenario: 1 August 2008 at 390 K (case 1) and 6 September 2008 at 390 K (case 2). We can sample the objective function for the given days Fig. 1 and Fig. 2 respectively.

For case 1 the objective function is low for small  $\mu_b$  which means that the sum of all values inside of the ASMA box are comparable to the sum of all values outside of the ASMA box. But eventually, when we increase  $\mu_b$  (e.g., subtracting higher value from the given MSF grid data) the values outside of the ASMA box converge to zero more rapidly than the values inside of the ASMA box and the objective function rise (Fig. 1). At some point we reach such  $\mu_b$  that will give us the highest sum of residual MSF values inside of the ASMA box while keeping the values outside of the ASMA box near zero. This  $\mu_b$  value gives us a position of the maximum of the objective function. Increasing further  $\mu_b$  will just decrease the values inside of the ASMA box (the values outside of the ASMA box are already zero) so the objective function starts to decrease and reaches zero.

## The objective function f, 01 August 2008 12:00 UTC at 390 K

Figure 1. The objective function  $f(\mu_b \mid \hat{\mu})$  depending on the background value  $\mu_b$  for 1 August 2008 at 390 K. The maximum around  $\tilde{\mu}_b \approx 366.4$  is the optimized background value. For this date, the majority of MSF values outside of the box are smaller than inside.

Figure 2. The objective function  $f(\mu_b \mid \hat{\mu})$  depending on the background value  $\mu_b$  for 6 September 2008 at 390 K with high MSF values observed outside of the box. The maximum around  $\tilde{\mu}_b \approx 365.7$  is the optimized background value.

For case 2, when a strong circulation outside of the ASMA box is present, the same steps described for case 1 remain true with the exception that when we reach some relative high  $\mu_b$  the values outside of the ASMA box do not become zero. This is why the objective function has a different shape but still the global maximum points at such  $\mu_b$  that will keep the sum of all residual MSF values inside of the box highest while the sum of residual MSF values outside of the ASMA box will be the lowest.

Both cases allows us to find the optimized background value  $\tilde{\mu}_b$  that will provide a contour line that will encircle the ASMA. After  $\tilde{\mu}_b$  is found, the absolute vortex moment (Eq. 2) can be used to define the ASMA boundaries as well as centroid position, the angle between the equatorial axis and the major axis of the ASMA, aspect ratio and excess kurtosis (Appendix A) for each point in time. Sometimes even  $\tilde{\mu}_b$  preserves small non-zero residual values (i.e., noise) outside of the box or another circulation

is also present as was noted before, so to calculate the absolute vortex moment quantities only values inside of the ASMA box are considered. With this approach, we can exclude small noise outside the ASMA box or the impact of other monsoon systems.

At this point we can retrieve time-series of the moments quantities. First the  $\tilde{\mu}_b$  value is determined for each day at 12:00 UTC for the period from 1980 to 2023 and then all moments quantities are calculated following the description (see Appendix A). Our approach is fundamentally different compared to Manney et al. (2021) where only one single background value  $\mu_b$  is used for each isentropic surface over the analysed time period from 1979 to 2018.

With our approach we capture more ASMA boundaries than Manney et al. 2021. The disadvantage of using a single background value for all points in time can be shown using Eq. 2. Two cases might occur:  $\min |\hat{\mu}_{i,j}| \gg \mu_b$  and  $\max |\hat{\mu}_{i,j}| \lesssim \mu_b$ . The operation

$$R_{i,j} := \hat{\mu}_{i,j} - \mu_b$$
, for  $i = 0, \dots, n-1$  and  $j = 0, \dots, m-1$  (6)

in Eq. 2 subtracts the background value  $\mu_b$  from each grid point. In the first case, the background value  $\mu_b$  might be not large enough to yield a boundary at all, in the second case the background value might cause the loss of information about optimized boundaries or even no information about the anticyclone at all.

In our approach,  $\widetilde{\mu}_b$  is calculated for each point in time that yields optimized boundaries of the ASMA during the entire Asian monsoon season. Manney et al. (2021) use a single background value per vertical level for the whole period analysed by Santee et al. (2017). Santee et al. (2017) took Montgomery streamfunction and wind speed values over a predefined region for June of a set of years and calculated a linear fit. The relationship then was used to retrieve MSF values that correspond to a chosen wind speed level (10 m s<sup>-1</sup> at 370 K and 390 K and 5 m s<sup>-1</sup> at 350 and 410 K) on each isentrope ( $\mu_b = 344.8, 356.5,$ 367.1, 377.3 ( $\times 10^3$  m<sup>2</sup> s<sup>-2</sup>) for 350, 370, 390 and 410 K respectively). To compare our method we used the Santee et al. (2017) methodology, but calculated  $\mu_b$  for each time-point for each vertical level. The comparison to our approach (resulting in  $\widetilde{\mu}_b$ ) is shown on Fig. 3 where we choose to show the Asian monsoon season 2023 (the other 43 years show similar time-series). During the developing (May-June) and break-up (September) phases of the anticyclone, the Santee et al. (2017) method gives lower values for  $\mu_h$  than our method, which means that some unwanted noise is still preserved when employing the Santee et al. (2017) methodology (note that Manney et al. (2021) use the background value  $\mu_b$  provided by Santee et al. (2017)). During the peak phase of the anticyclone the results of both methods are reversed: the Santee et al. (2017) approach gives higher values of the background value  $\mu_b$  than the value determined in this work, which leads to loosing some information on the edge of the anticyclone and shrinks its area at 370, 390 and 410 K. Our method shows  $\tilde{\mu}_b$  values for 350 K that have a small variation during the whole Asian monsoon season reflecting that at 350 K the ASMA is weak and capturing it poses a challenge. In contrast to the Santee et al. (2017) method that show an increase during the development of the ASMA and an decrease during the break-up. Our results are consistent with other methods such as the vorticity-based determination of the ASMA boundary that also found the ASMA at levels higher than 350 K (Ploeger et al. 2015).

**Figure 3.** The Montgomery streamfunction optimized background values using our approach (blue line)  $\widetilde{\mu}_b$  and according to Santee et al. (2017) (green dashed line)  $\mu_b$  at 350, 370, 390 and 410 K for the Asian monsoon season 2023.

The next step is to define the peak phase of the Asian monsoon anticyclone (referred to as "ASMA peak phase") to highlight the period of the anticyclone between the developing and break-up phases. In previous studies, different approaches were used to define the monsoon peak phase, very often a fixed time range was used, for example June-July-August (JJA) or July-August (JA) (e.g., Chen et al. 2021, Park et al. 2008).

Manney et al. (2021) use the methodology that is based on the area threshold. The ASMA peak phase is considered to begin (to end) when the anticyclone's area rises above (drops below) 1 % of the area of the hemisphere for 20 consecutive days before (after) the start (end) date.

Instead, we perform principal component analysis (PCA) over a set of quantities: pressure, PV, optimized background values  $\tilde{\mu}_b$  and ASMA area. The pressure and potential vorticity were averaged in the ASMA box to obtain a single value per time-step. All quantities were normalized before performing PCA – by retrieving the standard score (score =  $\frac{x-\bar{x}}{\sigma}$ ) for the period of 1980-2023. The ASMA peak phase is defined to start (end) when the first principal component is greater (less) than zero for 14 consecutive days.

To ensure that the ASMA box position is robust to adjustment we apply a sensitivity analysis by comparing a set of optimized background values of altered ASMA box position to the reference  $\widetilde{\mu_b}$  (see Appendix B).

The proposed methodology of encircling the anticyclone allows us to perform the interannual analysis using the centroid latitude and longitude, excess kurtosis, angle, aspect ratio and the ASMA relative area. As was mentioned before the anticyclone's area was not calculated via absolute vortex moments which provides the equivalent area of the ellipse. Instead we calculate the area (km²) of a polygon retrieved from the ASMA boundaries that is projected on the Earth surface and then take it as a fraction of the Northern hemisphere.

#### 240 3 Variability of the Asian summer monsoon anticyclone

#### 3.1 Horizontal distribution of the ASMA

The difference  $R_{i,j}$  (Eq. 6) between the Montgomery streamfunction grid values ( $\hat{\mu}_{i,j}$ ) and the optimized background  $\tilde{\mu}_b$  indicates the location of the ASMA (shown in Fig. 4) and is referred to as residual MSF. Figure 4 shows the mean residual MSF for 1980 to 2023 derived from 44 years of ERA5 1° × 1° reanalysis data from which the ASMA boundaries are inferred and centroid position for 370 K and 390 K isentropic surfaces (350 K and 410 K are shown in Fig. C1 in the Appendix) for each month from May to September. Each row consists of individual months and the columns represent the vertical level. The mean residual MSF does not represent the optimized boundaries of the ASMA at any specific time-point and are used here only for a qualitative description of the ASMA development from pre-monsoon to post-monsoon.

Figure 4 shows that during May the ASMA starts to develop and settles over South Asia at 370, 390 and 410 K (shown in Fig. C1 in the Appendix). At 350 K (Fig. C1 in the Appendix), only a very weak anticyclone is indicated and temporarily separated into two parts demonstrating that the main ASMA is above 350 K.

Mean residual MSF values rise and fall through May-September and peak in July-August. Further, the position of the anticyclone changes during the ASMA peak phase. A northward shift of the ASMA centre is found at the beginning of the Asian summer monsoon and a southward shift is found at the end of the monsoon season in agreement with previous studies (e.g., Vogel et al. 2015; Goswami 2012).

From Fig. 4 we can see that starting in June the values tend to cluster around both sides relative to the centroid position, indicating a western part (Iranian mode) and eastern part (Tibetan Mode) of the anticyclone.

During July stronger values of the mean normalised MSF were found in the western part compared to the eastern part, even the area of the western part is larger in the climatological mean. This feature suggests a possible bimodality of the ASMA which is discussed in more detail in Sec. 3.2.

During break-up phase in September, the shape of the mean residual MSF is elongated and shrunk along latitude axis, indicating the break-up of the anticyclonic circulation. The temporal evolution of the ASMA during the monsoon season will be discussed in Sec. 3.3.

Figures 5, 6 and 7 show examples at 390 K of the ASMA boundaries, potential vorticity, wind velocity and corresponding moments quantities for three days during which relatively high absolute values of the angle and excess kurtosis were deduced. During 15 July 2017 at 390 K the anticyclone was titled counter-clockwise and the angle quantity is  $-7.04^{\circ}$ , which can be observed in the corresponding time-series (Fig. 12). Figure 6 shows a positive angle (1.83°) because of the boundary curvature

**Figure 4.** Mean residual MSF for 1980-2023 for the Asian monsoon anticyclone (the difference between MSF values and the background value) for each individual month from May to September (rows) at 370 K and 390 K (columns).

to the north around 60° E. Fig 7 shows a splitting-like behaviour of the anticyclone and how the excess kurtosis is enhanced (1.5) compared to the example on Fig. 5 where no splitting occurs.

In addition we provide a comparison of the boundary determined with our method and the Ploeger et al. (2015) PV-based transport barrier of the ASMA. Figure 8 shows the PV-based barrier as a red dashed line and our boundary is denoted as a black line for 14 July 2022 at 370 K. Both methods capture the curvature of the northern side of the boundary around 80° E. The southern part is also aligned for the two methods and both follow wind velocity arrows. The Ploeger et al. (2015) boundary

of the ASMA contains somewhat more noise compared to the boundary in this work. In contrast to Ploeger et al. (2015) our method suggests a western tail of the ASMA between  $0^{\circ}E - 10^{\circ}E$ . Using the Ploeger et al. (2015) method it is not always possible to determine a PV barrier on a daily scale, for this reason interpolated PV barrier is taken for comparison and illustrated in Fig. C2. While our method captures the boundary around  $0^{\circ}E - 135^{\circ}E$  and the wind velocity aligns with the deduced ASMA boundary, the interpolated PV boundary is only able to encircle the ASMA in the range  $90^{\circ}E - 135^{\circ}E$ .

**Figure 5.** Horizontal plot of the ASMA boundaries (thick black line), potential vorticity (colour map), wind velocity (arrows) and corresponding moments quantities for 15 July 2017 at 390 K.

**Figure 6.** Horizontal plot of the ASMA boundaries (thick black line), potential vorticity (colour map), wind velocity (arrows) and corresponding moments quantities for 18 July 2022 at 390 K.

#### PV, ASMA boundaries (black) and Wind velocity (white arrows) for 25 June 2022 at 390 K 12.0 80°N 10.5 Area: 8.1% Longitude: 68.4 [deg] Potential Vorticity [PVU] 70°N 9.0 Latitude: 32.1 [deg] Excess kurtosis: 1.5 60°N 7.5 Anale: -2.09 [dea] Aspect ratio: 4.37 50°N 6.0 40°N 4.5 30°N 3.0 20°N 1.5 0.0 ٥° 25°E 50°E 75°E 100°E 125°E 150°E 175°E

**Figure 7.** Horizontal plot of the ASMA boundaries (thick black line), potential vorticity (colour map), wind velocity (arrows) and corresponding moments quantities for 25 July 2022 at 390 K.

**Figure 8.** Horizontal plot of the ASMA boundaries (thick black line), potential vorticity colour map (PV\_MEAN as defined in Ploeger et al. (2015)), wind velocity (arrows), PV barrier (red dashed line) and corresponding moments quantities for 14 July 2022 at 370 K.

#### 3.2 Bimodality and eddy shedding of the ASMA

Nützel et al. (2016) used seven reanalyses and found strong evidence of bimodality only in the outdated reanalysis NCEP-R1 and no evidence in daily data (or limited evidence for monthly data) in more modern reanalyses. They identified the center of the ASMA using the data of the absolute zonal wind field and the geopotential field at 100 hPa (Zhang et al. 2002). Manney

**Figure 9.** Mean JJA Hovmöller diagram of normalized zonal mean  $(15^{\circ} N - 45^{\circ} N)$  of the residual MSF for 1980-2023 period at 370 K and 390 K. Bright colours indicate where the maximum residual MSF values tend to cluster.

et al. (2021) used the MERRA-2, ERA-I and JRA-55 reanalyses and support the results of Nützel et al. (2016) for recent reanalyses. The recent study of Siu and Bowman (2020) showed that ASMA contains two or three simultaneous subvortices using the ERA-Interim dataset.

The temporal variability of the bimodality is analysed using Hovmöller diagrams at 370 K and 390 K (Fig. 9; 350 K and 410 K are shown in Fig. E1 in the Appendix). A zonal mean  $R_{i,j}$  (15° N – 45° N) of the residual MSF for JJA of each year (1980-2023) (left) with corresponding optimized background values (right) are shown. High values of the zonal mean of the residual MSF indicate the longitudinal position of the ASMA. Because we are interested only in the positions, the  $R_{i,j}$  values were normalized individually by dividing each year separately using its own maximum value. This way we can qualitatively analyse the spatio-temporal distribution of residual MSF clusters and analyze a possible bimodality. For example, in 1981 the

**Figure 10.** Number of days per year when normalized residual zonal mean  $R_{i,j}$  is above the threshold (> 0.5) in both the western and eastern part at 370 K (top) and 390 K (bottom) indicating bimodality i.e., the simultaneous existence of two parts (modes) of the ASMA. The separation line that splits the western and eastern part is defined as average centroid longitude of the ASMA for the corresponding year.

peak values are clustered around  $45^{\circ}$ E and simultaneously another cluster further to the east around  $125^{\circ}$ E is found (Fig. 9). A similar behaviour we can see in 1986 but the clusters splits closer to around  $80^{\circ}$ E with the east part being weaker but still strong in terms of residual MSF having  $\approx 85\%$  values of the maximum. One of the noticeable difference between the years is that during the 80s there were fewer splitting events than in the recent years.

To infer if the western and eastern part (modes) of the ASMA occur simultaneously or alternating (or both) we analyse the zonal mean residual MSF on a daily basis. The Hovmöller diagrams shows the possible bimodality of the ASMA in JJA, in addition we can quantify daily splitting of the residual MSF. Figure 10 shows the number of days per year when the normalized zonal  $R_{i,j}$  is above the threshold (50% of the max value as a reasonable threshold to check where the residual MSF values at least as half of the maximum are placed) in both parts of the divided longitudinal axis – the west (all positions < mean centroid longitude) and the east (all positions > mean centroid longitude) parts. We calculate the average centroid longitude of

the ASMA for each year and it serves as a logical line to divide the axis in two parts. If for each day the  $R_{i,j}$  values above the threshold occurred in the both parts we count it as a bimodal day.

In Fig. 10 we can see that for the majority of the years the bimodality of the ASMA usually lasts for less than 10 days for June-August period at 370 K, however a larger number of days is found at 390 K (up to 40 days).

We presented three types of plots to show possible bimodality of the ASMA: climatological horizontal plots for  $R_{i,j}$ , Hovmöller diagrams of zonal  $R_{i,j}$  during JJA for each year and number of days per year when zonal  $R_{i,j}$  is above the threshold simultaneously in multiple positions. All of them show that there is a bimodality of the ASMA if we use our methodology to define the optimized background and residual MSF.

### 310 3.3 Intraseasonal and interseasonal variability of the Asian summer monsoon anticyclone

Figures 11, 12 and 13 show the intraseasonal variability of the climatological mean (1980-2023) of the anticyclone's moments, namely the centroid position of the ASMA latitude and longitude, excess kurtosis, the angle and aspect ratio and, additionally, the relative area at 370 K and 390 K (350 K and 410 K are shown in Figs. D1, D2 and D3 in the Appendix) from May to September. Moreover, we show individual time-series for 2017, 2022 and 2023 to highlight years when aircraft measurements inside the ASMA were conducted during the StratoClim (Stroh and StratoClim-Team 2025), the ACCLIP (Pan et al. 2024) and PHILEAS (Riese et al. 2025) aircraft campaigns respectively.

Three years are highlighted when measurement campaigns took place to show that it is possible to provide a comprehensive analysis on a finer time-scale without the need to fix one specific constant value of the MSF. This might help the scientific community to thoroughly analyse the results of those campaigns in future work. This is also the motivation why a method is needed that works on any time-scale without depending on averaging the data.

A strong variability are found in the centroid position of the ASMA latitude and longitude for the individual years 2017, 2022 and 2023 (Fig. 11) with an increased variability of the latter two years. In all three years time periods exist in which the variability of the centroid position of the ASMA is lower or greater than the standard deviation of the climatological data.

In particular during August 2022, there is a strong shift of the ASMA to the east compared to the climatological mean data. Therefore aircraft flights inside the ASMA could be performed during the ACCLIP campaign over South Korea (Pan et al. 2024) contrary to the original campaign goal to measure the eastern outflow of the ASMA.

But also during the PHILEAS campaign (Riese et al. 2025) to measure the eastern outflow of the ASMA over Alaska, there was a strong shift of the ASMA to the east compared to the climatological mean data during the first half of both August and September 2023.

The intraseasonal variability of the centroid latitude position indicates a geographical northward shift of the ASMA until August and a subsequent southward shift towards the tropics afterwards. The centroid latitude time-series (Fig. 11, left) of the anticyclone is positioned between  $10^{\circ} N - 20^{\circ} N$  in May at 370 K and 390 K. During the ASMA peak phase the centroid position moves northward up until August and then starts to move to the south again. In September the centroid position is placed further northward than when it started near  $20^{\circ} N - 25^{\circ} N$ . The northward movement of the ASMA is further to the north during 2022 than for 2023 for almost the whole May-September time-series at all vertical levels although the difference

CENTROID LATITUDE CENTROID LONGITUDE 370 K 370 K 40 140 35 120 30 Lat [deg] Lon [deg] 80 20 60 40 15 390 K 390 K 40 140 35 120 30 Lon [deg]

80

60

1 Jul

2017

1 Aua

2022

1 lun

Mean 1980-2023

1 Oct

20

340

345

1 Jul

2017

1 lun

Mean 1980-2023

1 Aug

1 Sep

2023

Figure 11. Climatological (1980-2023) time-series (black line) of the centroid position of the ASMA latitude (left hand side) and longitude (right hand side), and individual years of 2017 (yellow dots), 2022 (blue downward-pointing triangle) and 2023 (red plus) at 370 K and 390 K. The lavender background denotes the standard deviation of the climatological data.

1 Oct

is small, only quickly at the beginning of August longitude centroid positions for both 2022 and 2023 were nearly the same. The year of 2017 usually better aligns with the climatology than 2022 and 2023.

The intraseasonal variability of the centroid longitude position indicates a geographical westward shift of the ASMA until end of July and a subsequent eastward shift afterwards. The centroid longitude time-series (Fig. 11, right) of the anticyclone shows the evolution of the position start on the east near  $150^{\circ}$  E, eventually shifting to the west where it occupies the longitudinal band around 70° E at 370 K and 390 K. After its peak, the position of the ASMA centroid is shifted to the east during the end of the monsoon season and returns to the location of 150° E in September. The difference of the latitude and longitude centroid position for the years 2017, 2022 and 2023 and the climatological mean indicates the strong interseasonal variability of the ASMA. Looking into highlighted years in June the position of the ASMA centroid longitude is shifted to the east in 2017 and 2023 more than in 2022 at all vertical levels with a maximum difference approximately 30°. The difference between the latitude centroid position for the years 2017, 2022 and 2023 and the climatological mean indicates the strong interseasonal variability of the ASMA.

**Figure 12.** Climatological (1980-2023) time-series (black line) of the ASMA area (left hand side) and angle (right hand side), and individual years of 2017 (yellow dots), 2022 (blue downward-pointing triangle) and 2023 (red plus) at 370 K and 390 K. The lavender background denotes the standard deviation of the climatological data.

The years 2017, 2022 and 2023 highlight the periodic nature of the ASMA area; during June-August south-northward as well as east-westward oscillations are found, similar as oscillations detected in Asian summer monsoon rainfall (Goswami 350 2012).

The area of the ASMA (Fig. 12, left) is increasing during its development until mid-June, stays nearly constant in its peakphase and starts to decrease end of August. In May the area of the ASMA is only about 2% of the area of the hemisphere and then gradually starts to increase. The steepest rise is seen at 370 K during mid of May. During the ASMA peak phase between June and September the ASMA area stays roughly the same at level of 10% of the hemisphere at 370, 390 and 410 K. At the beginning of September the area starts to decrease at 370 K and 390 K, for 410 K the process starts earlier – in August.

The highlighted years (2017, 2022, 2023) generally tend to group around the climatology. The values of the time-series are more robust at 370 and 390 K (Fig. 12) and the majority of the ASMA areas lie within one standard deviation of the climatology in contrast to 350 and 410 K (Fig. D2, left).

**Figure 13.** Climatological (1980-2023) time-series (black line) of the ASMA excess kurtosis (left hand side) and aspect ratio (right hand side), and individual years of 2017 (yellow dots), 2022 (blue downward-pointing triangle) and 2023 (red plus) at 370 K and 390 K. The lavender background denotes the standard deviation of the climatological data.

These oscillations are averaged out when considering the climatological time-series but a quick decline of the ASMA area can be seen at the end of July at all vertical levels. This decline in the ASMA area values varies between altitude, at 350 K the ASMA area decreases almost to zero for all highlighted years. During 2022 and 2023 the ASMA area cuts roughly by half from its climatological level at 370, 390 and 410 K.

The temporal evolution of the climatology of angles between the equatorial axis and the major axis of the ASMA ellipse are very low (around zero) indicating that the major axis of the ASMA is almost parallel to the equator during the monsoon season (Fig. 12, right). The individual years in contrast show low angle oscillations during the whole season. They have something similar to a phase shift relative to each other and that is why the climatological data are almost always equal to zero.

One relatively high deviation from zero can be seen in July 2022 at 390 K and 410 K when the angle decreases below zero at first but then rises above zero before stabilising. A similar event was not found in 2017 and 2023.

Excess kurtosis (EK) measures the degree of splitting of the ellipsoid and therefore can serve as a metric for the bimodality (trimodality) of the ASMA (Fig. 13). During the developing phase of the ASMA up to until June and also during the break up

phase after September the standard deviation of EK (and thus its variability) is higher than in the ASMA peak phase during the June-September period at all vertical levels and cannot be a reliable description of the ASMA evolution. The peaks of EK time-series when the ASMA experiencing the split are usually filtered out from the climatological values during averaging and they are better represented in individual years. The temporal evolution of the EK during 2022 shows that in June-September period there are more splitting like events than in 2023 at 390 K.

The aspect ratio (Fig. 13, right-hand side) is the ratio between major and minor axis of the ellipse and quantifies its elongation. During the ASMA peak phase the aspect ratio is between  $\sim$ 1 and 10 at all vertical levels. Progressing through the season both the climatology and individual years show a good alignment of slowly increasing aspect ratio indicating that the anticyclone is stretching with time in west-east longitudinal direction. Comparing individual years it can be noted that between July and August there are a couple of rapid elongations for 2017 and 2023 at 370, 390 and 410 K. For 2022 there is only one rapid increase of the aspect ratio during late June at all vertical levels except 350 K.

Figures 11,12 and 13 show a good overall agreement with the results of the study by Manney et al. (2021). Our method gives us an indication of the ASMA's evolution between May and September despite a relatively strong variability. Both centroid latitude and longitude and the aspect ratio show similar temporal evolution as the Manney et al. (2021) findings. Although the area of the ASMA occupies around 10% of the hemisphere in the beginning of August in our results and Manney et al. (2021) (MERRA-2), the shape of the time-series is more flat in this work during the peak phase of the ASMA compared to Manney et al. (2021). Because we provide individual campaign years in addition to climatological time-series, our results show oscillations of excess kurtosis and angle moments compared to Manney et al. (2021), who focus on the climatological feature of the ASMA behaviour.

### 3.4 Duration of the ASMA peak phase

To determine the start and end dates of the ASMA peak phase, we use the similar approach as Manney et al. (2021) (the ASMA peak phase starts (end) when the area is greater (lower) than 1% of the hemisphere for 20 consecutive days). But instead, PCA technique was used to obtain the index time-series (see Sec. 2). The ASMA peak phase is defined to start (end) when the index is greater (less) than zero for 14 consecutive days. Fig. 14 shows the time development of the start dates, end dates and the duration of the peak phase of the Asian summer monsoon anticyclone at three different levels of potential temperature. Each row on Fig. 14 denotes the vertical level and each column is for the anticyclone start, end and the duration in days (end - start + 1) respectively.

Manney et al. (2021) show that the anticyclone's peak phase tends (1979-2018) toward earlier formation, later break-up and longer duration at 350, 370, 390 and 410 K, with much larger trends at the lower levels. Our calculations (Fig. 14) show similar results at 370 K where we found largest trends (we do not provide the data for 350 K because this level is below the main anticyclone). There are mixed results at higher altitudes where end dates show negative trend at 390 K and positive trend at 410 K. Overall, the duration is sensitive to initial conditions (e.g., the quantity time-series or the threshold). Manney et al. (2021) show that the slopes of the duration also vary with respect to the reanalysis (MERRA-2, ERA-I or JRA-55).

**Figure 14.** Start (left), end (middle) dates and the duration in days (right) of the ASMA peak phase at 370, 390 and 410 K. The red triangle denotes the mean value. Colour marks ENSO type of a year based on ONI index (using DJF, from the winter before the considered monsoon period). The year is marked as El Niño (red) if the index is  $> 0.5^{\circ}C$ , La Niña (blue) if the index is  $

**Figure 15.** Trends of the ASMA area, centroid latitude and longitude for JJA based on ERA5 reanalysis. Colour marks ENSO type of a year based on ONI index (using DJF, from the winter before the considered monsoon period). The year is marked as El Niño (red) if the index is  $> 0.5^{\circ} C$ , La Niña (blue) if the index is  $

Figure 16. Same as Fig. 15 (left), but using the methodology by Manney et al. (2021).

425

optimized one based on the introduced objective function it leads to the ASMA area decreasing over the same period. If we look at the area time-series (Fig. 12, left), we notice that during June-August the area is almost flat in contrast to the findings by Manney et al. (2021). The ASMA area in JJA is averaged for each year and shown on Figure 15. It seems to give us an information that the area is decreasing. Manney et al. (2021) use a single background value per vertical level, hence the increase in area might be because of the spread of the higher MSF values in the region and the extent of area increase depends on the chosen fixed MSF value. Our findings compared to Manney et al. (2021) show that the calculation of the change in the ASMA area over time depends in detail on the used methodology.

During El Niño years, the ASMA centroid is somewhat shifted towards the east and to the tropics and ASMA area size seems not to be correlated to ENSO at 370, 390 and 410 K. Manney et al. (2021) report similar conclusions for correlation between MEI (Multivariate ENSO Index) and ASMA area and centroid position.

#### 4 Discussion

440

450

In the following we will link our findings to results of several previous studies to discuss possible similarities and inconsistencies. The downward trend of the ASMA area and its bimodality deduced here aligns with multi-decadal weakening (1958-2020) reported by Qie et al. (2025), who concluded that changes in wave activity impact the strength of the ASMA.

Siu and Bowman (2020) developed an algorithm that allows multiple simultaneous subvortices of the ASMA to be tracked. They used the ERA-Interim reanalysis and showed that the ASMA contains two or three distinct subvortices 69 % of the time simultaneously. Siu and Bowman (2020) show that in many cases there are two broad peaks of the ASMA – between about 40° E and 100° E and near 150° E.

Our results are similar to Siu and Bowman (2020) but in contrast to the latter study we work with residual MSF on isentropic surface and consider first climatological horizontal plots to see qualitatively the existence of multiple peaks at 370, 390 and 410 K between June and September (Fig. 4; Fig. C1); the Hovmöller diagram of normalised zonal mean residual MSF values for each year to see the distribution of peaks and count number of days when there is a simultaneous presence of two peaks on both sides relative to the longitudinal centroid separation line. Our results show that there is a clustering of peaks during 1980-2023 near 50° E and near 150° E (for some years) that indicates a bimodal state of the ASMA similar to the findings by Siu and Bowman (2020).

The sharpening of trends and the bimodality of the ASMA also resonates with the jet-referenced geopotential-height (GPH) method of Musaid et al. (2024) which analyses active (widespread rainfall is observed over central India) and break (primarily dry conditions prevail throughout central India) monsoon phases by referencing the tropical easterly jet and subtropical westerly jet and GPH. Although the authors do not provide a quantitative analysis of the spatial evolution of the ASMA they claim that the constant GPH method provides "unrealistic" stretch of the ASMA compared to their method during active and break phases. In our analysis the aspect ratio time-series shows that the ASMA starts to stretch starting in June and becomes the widest along longitudinal axis in October by almost a factor 1.8 relative to the beginning of June. Similar positive trends of the climatological aspect ratio are also shown by Manney et al. (2021).

#### 5 Conclusions

We introduced a novel method based on the absolute vortex moments that is used to define boundaries of the Asian summer monsoon anticyclone. This method allows the interannual variability of the anticyclone and possible trends to be analysed. The method makes it possible to describe the ASMA on individual days from its formation phase in May-June, during the peak phase to its break-up phase in September. In this work, the most recent ECMWF reanalysis ERA5 is used in contrast to

previous studies (e.g., Manney et al. 2021) using earlier reanalyses (MERRA-2, ERA-Interim, JRA-55) having lower spacial resolution. The ASMA was analysed over a time period covering 44 years (1980-2023) on selected isentropic surfaces (370 K and 390 K; 350 K and 410 K are shown in Appendix C). In addition to climatological mean values over 44 years we investigate the anticyclone's behaviour during individual years – 2017, 2022 and 2023 – when the aircraft campaigns StratoClim, ACCLIP and PHILEAS were conducted, respectively, in the region of the ASMA.

Our method shows good overall agreement with studies employing a PV-based boundary of the monsoon anticyclone but can be applied to individual points in time. Using the method we were able to identify bimodality (both on daily and seasonal time scales) of the ASMA.

Our analysis shows the qualitative spatial evolution of the ASMA on fixed levels of potential temperature (350, 370, 390 and 410 K). The distribution of the difference  $R_{i,j}$  between the Montgomery streamfunction grid values ( $\hat{\mu}_{i,j}$ ) and the optimized background  $\tilde{\mu}_b$  highlights climatological regions where the ASMA is occurring during corresponding months. At 370, 390 and 410 K the distribution is similar. When the anticyclone starts to form in May all vertical levels show a higher aspect ratio than during the peak phase. We show that horizontal plots of the residual MSF values provide the information about splitting-like behaviour of the anticyclone in July, August and September supporting the recent study of Siu and Bowman (2020) who showed multiple simultaneous peaks of the ASMA. With our new method and using the ERA5 reanalysis, we can confirm a bimodality of the ASMA.

The moments analysis provides a quantitative method based on six quantities using which we build spatio-temporal characteristics of the ASMA: the centroid latitude and longitude, excess kurtosis, angle, aspect ratio and area.

The time-series show that:

- The area of the ASMA increases gradually during May-June at 370, 390 and 410 K with the steepest rise seen at 370 K during mid of May. Individual years show an essential area drop at the beginning of August.
- South-north shift of the centroid position of the ASMA oscillates during the season and tends to shift poleward with vertical level. Individual years when campaigns were performed (2017, 2022 and 2023) are close to the climatology but the anticyclone tends to reside more northward during 2022 than in 2023.
  - East-west shift evolves during the season, it has a minimum near 50° E at all vertical levels. The time-series of the centroid longitude starts and ends around 100° E. There is a significant difference between 2022 and 2023 in end of May and beginning of June. The ASMA is shifted more to the east during 2022 and the difference is around 30° compared to 2023 at all vertical levels.
  - The excess kurtosis of the ASMA is near zero for the climatology but has distinct spikes in individual years indicating a splitting-like behaviour showing bimodality and shedding events.
  - On average the angle of the ASMA relative to the equator is near zero, but individual years oscillate and have shifted relative to each other.

- The aspect ratio of the ASMA has higher standard deviation during formation and break-up phase indicating its instability. During the whole season the values show a linear growth of the aspect ratio at all vertical levels. There is a shrinking and elongation effect that can be seen in individual years, especially in 2023.

Using the novel method, we determine start and end dates of the ASMA and thus its duration in days. The starting dates for the ASMA peak phase are between May-June. The end dates are in between September and October at 370, 390 and 410 K. The overall duration of the ASMA peak phase is around 120 days and has increased over the analysed time period. The slopes of the trend lines suggest that the ASMA peak phase tends to start earlier and end later especially at lower altitudes over the period from 1980 to 2023 the monsoon period extended by  $\sim 10$  (at 410 K) and  $\sim 40$  (at 370 K) days. These findings are consistent with previous studies (Manney et al. 2021).

As long as the ASMA exists, it confines air masses with tropospheric characteristics (e.g., pollutants) with the possibility of east- and west-ward export by eddy-shedding as well as further transport higher up into the stratosphere. The duration of the ASMA is thus an important quantity.

Interannual trends for 1980-2023 were determined for the anticyclone's area, centroid latitude and longitude. The area shows a decline over this period for all vertical levels. This results contradicts previous studies and is caused by the method itself; an impact of the used reanalysis could be excluded. Our findings show that the ASMA centroid position depends on ENSO. During El Niño (La Niña) years, the ASMA centroid is somewhat shifted towards the east (west) and to the tropics (mid-latitudes). This issue needs to be analysed in more depth in further studies.

Overall, the new proposed method shows a straightforward way to calculate the moments quantities with reduced noise. The chosen methodology allowed us to conduct the robust analysis which gave us similar results with previous studies for centroid position, aspect ratio seasonal time-series and start and end dates of the ASMA. In addition our method highlighted differences in the interannual changes of the ASMA area that decreases compared to Manney et al. (2021) and allowed a possible bimodality of the ASMA to be analysed. Our new approach to describe the ASMA and its boundaries can be used on any time scale, choosing individual days or specific hours as well as climatological time scales of several decades to analyse previous or future possible changes of the Asian monsoon anticyclone caused by climate change.

#### 515 Appendix A: The quantities retrieved using the absolute vortex moment

Using Eq. (3) Matthewman et al. (2009) define the equivalent ellipse by its centroid position  $(\bar{x}, \bar{y})$ , its angle of orientation  $\psi$ , its excess kurtosis  $\kappa_4$  and its aspect ratio r:

$$(\bar{x}, \bar{y}) = \frac{1}{M_{00}}(M_{10}, M_{01}).$$
 (A1)

To define  $\psi$  and r Matthewman et al. (2009) introduce the relative vortex moment:

$$J_{kl} = \int \int [\hat{\mu}_{i,j} - \mu_b] (x - \bar{x})^k (y - \bar{y})^l dx dy,$$
 (A2)

then

$$\psi = \frac{1}{2} \tan^{-1} \left( \frac{2J_{11}}{J_{20} - J_{02}} \right),\tag{A3}$$

$$r = \left| \frac{(J_{20} + J_{02}) + \sqrt{4J_{11}^2 + (J_{20} - J_{02})^2}}{(J_{20} + J_{02}) - \sqrt{4J_{11}^2 + (J_{20} - J_{02})^2}} \right|^{1/2},$$
(A4)

and

525 
$$\kappa_4 = M_{00} \frac{J_{40} + 2J_{22} + J_{04}}{(J_{20} + J_{02})^2} - \frac{2}{3} \left[ \frac{3r^4 + 2r^2 + 3}{(r^2 + 1)^2} \right].$$
(A5)

#### Appendix B: Sensitivity analysis

As was mentioned earlier, the ASMA box was defined as a region of ASMA occurrence and its position was introduced on our empirical knowledge where the Asian summer monsoon anticyclone can be found. To ensure that our approach is robust to adjustment of the latitude-longitude box position we apply a sensitivity analysis. The strategy is as follows: using Eq. 4 we consider one specific  $\hat{\mu}_{i,j}$  but instead adjusting the ASMA box position by 1° starting from (0° N-90° N,0° E-180° E) until (-90° S-0° N,180° E-1° W) and retrieving  $\tilde{\mu}_b$  for each ASMA box. Figure B1 shows a distribution of the  $\tilde{\mu}_b$  with corresponding latitude-longitude shift.  $\Delta$ Lon,  $\Delta$ Lat axes show how many degrees are added to the ASMA box. As can be seen, the crossed region marks the range of step-sizes in degree that can be added to the ASMA box while preserving the original  $\tilde{\mu}_b$  (using the ASMA box from the methodology). The range of  $\Delta$ Lon,  $\Delta$ Lat are within 0 – 20 degree to the south and 0 – 40 degree to the east confirming that our approach is robust relative to a fluctuation of the position of the ASMA box.

## Lat-lon shift of the ASMA box and the corresponding objective function minimum Crossed region denotes values within

Figure B1. Sensitivity results of the latitude-longitude shift of the ASMA box and corresponding  $\tilde{\mu}_b$  using the MSF values from 1 August 2023 at 390 K. Crossed region highlights values within original background value  $\pm 0.01 \times 10^3$  m<sup>2</sup> s<sup>-2</sup>.

## Appendix C: Climatology of the horizontal distribution plots

The distribution  $R_{i,j}$  at 350 K has no information in May at 350 K telling that the anticyclone has only started to form and it's residual values are negligible on the chosen scale. Overall during the season  $R_{i,j}$  is small at 350 K compared to the rest of vertical levels. The values of  $R_{i,j}$  at 410 K are stronger also smaller than 370 K and 390 K.

**Figure C1.** Mean residual MSF for 1980-2023 for the Asian monsoon anticyclone for each individual month from May to September (rows) at 350 K and 410 K (columns).

## PV, ASMA boundaries (black), Wind velocity (white arrows) and PV barrier (red dashed line)

**Figure C2.** Horizontal plot of the ASMA boundaries (thick black line), potential vorticity colour map (PV\_MEAN as defined in Ploeger et al. (2015)), wind velocity (arrows), interpolated PV barrier (red dashed line) and corresponding moments quantities for 20 August 2022 at 370 K.

### 540 Appendix D: Climatological time-series of the ASMA moments quantities at 350 and 410 K

545

The moments quantities of the ASMA at  $350\,\mathrm{K}$  and  $410\,\mathrm{K}$  overall behave similar to  $370\,\mathrm{and}$   $390\,\mathrm{K}$  but the standard deviation at  $350\,\mathrm{K}$  and  $410\,\mathrm{K}$  are higher. Excess kurtosis which shows the splitting-like behaviour has more deviations at  $410\,\mathrm{K}$  than  $370\,\mathrm{K}$  and  $390\,\mathrm{K}$  and it does not serve as a reliable variable for  $350\,\mathrm{K}$ . Similar can be seen with the angle quantity where higher tilting is presented at  $350\,\mathrm{K}$  and  $410\,\mathrm{K}$  than  $370\,\mathrm{K}$  and  $390\,\mathrm{K}$ . The area of the ASMA is usually smaller at  $350\,\mathrm{K}$  and  $410\,\mathrm{K}$  and also have higher standard deviation.

**Figure D1.** Climatological (1980-2023) time-series (black line) of the ASMA centroid latitude (left hand side) and longitude (right hand side), and individual years of 2017 (yellow dots), 2022 (blue downward-pointing triangle) and 2023 (red plus) at 350 K and 410 K. The lavender background denotes one standard deviation of the climatological data.

**Figure D2.** Climatological (1980-2023) time-series (black line) of the ASMA area (left hand side) and angle (right hand side), and individual years of 2017 (yellow dots), 2022 (blue downward-pointing triangle) and 2023 (red plus) at 350 K and 410 K. The lavender background denotes one standard deviation of the climatological data.

**Figure D3.** Climatological (1980-2023) time-series (black line) of the ASMA excess kurtosis (left hand side) and aspect ratio (right hand side), and individual years of 2017 (yellow dots), 2022 (blue downward-pointing triangle) and 2023 (red plus) at 350 K and 410 K. The lavender background denotes the standard deviation of the climatological data.

## Appendix E: Bimodality of the ASMA

**Figure E1.** Mean JJA Hovmöller diagram of normalized zonal mean  $(15^{\circ} N - 45^{\circ} N)$  of the residual MSF for 1980-2023 period at 350 K and 410 K. Bright colours indicate where the maximum residual MSF values tend to cluster.

Author contributions. The study was conceived by OK, BV and GG. OK designed the method and conducted calculations for this study. BV directed the structure of the paper, analysed the calculated results and supported the study. GG provided advice and guidance for the analysis of the novel method. RM helped to better analyse the bimodality of the ASMA. The results were discussed by all the co-authors. The paper was written by OK with contributions from all the co-authors.

550

Competing interests. At least one of the (co-)authors is a member of the editorial board of Atmospheric Chemistry and Physics.

Acknowledgements. We thank the European Centre for Medium-Range Weather Forecasts (ECMWF) for providing the ERA5 reanalyses and the Jülich Supercomputing Centre (JSC; Research Centre Jülich, Germany) for the storage resources on the MeteoCloud data archive. The presented work were mainly funded by the German Science Foundation (Deutsche Forschungsgemeinschaft, DFG) as part of the HALO Priority Program SPP 1294 (project VO 1276/7-1). We would like to sincerely thank Michelle Santee (JPL, Pasadena) and Felix Ploeger (FZJ, Germany) for helpful discussions.

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
