# Peer review of "An optimization-based approach to track the Asian summer monsoon anticyclone across daily and interannual variability"

_EGUsphere, 2025_

## Referee Comment (RC2)

**Summary**

The authors investigates the trend and variability of Asian monsoon summer anticyclone (ASMA) characteristics using a novel method based on "absolute vortex moment", which can be used to describe various parameters of the ASMA such as its position, aspect ratio, among others. Given the disagreements in trends regarding ASMA strength (see Manney et al (2021) and Qie et al. (2025)), I think this study presented by Kachula et al. is a meaningful contribution as it presents yet another view on ASMA trends. In addition, the method presented here allows descriptions of vortex splitting, aspect ratio, and others, that are lacking in the literature. I think the scientific contribution of this work warrants publication after refinement of the manuscript.

**General comments**

- Siu and Bowman (2020) investigates the modality of ASMA and has some extensive discussions on the variability of ASMA having one, two, or more vortices. I notice that you do not refer to this work; I suggest looking at their research and adding some discussion on consistencies or differences between your work and theirs. In addition a recent work Qie et al. (2025) also investigated the trend of the ASMA. I think incorporated the finding of Qie et al. in their discussion and result interpretation.
- Throughout the interpretation of results, the three flight campaigns (StratoClim, ACCLIP, PHILEAS) are mentioned and the corresponding years are chosen for analysis, but your results for these years lack impact if we don't know what the campaign had found. Are any of the campaign's results explainable with your findings? If so I think they should be included in your discussions.

**Specific comments**

- It seems like one of the advantages of your method is its ability to capture complex shapes of the ASMA. For instance, in Figure 8 your method shows the tail to the west at 30N while the Ploeger method does not.
- Line 142: ASIA box is from 0E to 180E. Though this is not the norm but we often observe anticyclone stretching west of 0E. You can see this in your Figure 6. Is it possible to shift the box westward?
- Starting at line 196, the statement "During the developing (May-June)... the Santee et al. (2017) method gives lower values than our method, which means that some unwanted noise is still preserved... " Though Fig 3 does show lower values the Santee method, I think it should be explicitly shown (perhaps with some cases) that the method Santee et. al indeed performs worse. Likewise, for the anticyclone peak phase, the authors can show that their method defines the anticyclone better.
- Line 202 states that the ASMA at 350 K does not exist. I think this statement is not true. Figure 1 below is Montgomery streamfunction at 350 K and it is evident that a closed contour (i.e. anticyclone exists).

[Figure]

*Figure 1 350 K Montgomery streamfunction (black) for 2004/06/01 18Z*

- Line 250; Figure 5/6/7 – The "angle" quantity is discussed without much context. I think there needs to be some explanation on how to interpret this angle (perhaps a schematic or idealized cases) and what its implications are. For Figure 6, why is the angle related to the curvature at 60E but not the western side (e.g. 40E?).
- Figure 9/10 – Could you consider adding a subplot to show whether a JJA has 1, 2, or more maxima? I wouldn't demand this be done but think it can help us see whether a specific JJA tended to be bimodal or not.
- Line 346 – "The temporal evolution of the EK during 2022 shows that in June-September period there are more splitting like events than in 2023 at 390K." Do you infer this by seeing more higher values of blue dots verses red dots? Just asking for clarification.

**Minor comments**

- There are many short paragraphs, some with one sentence. I suggest revising the writing to reduce the number of these.
- Line 102 – Is this supposed to be 0.25 degrees?
- Figure 5, 6: I suggest changing the colormap. The choice of colormap depends on what you want to highlight (i.e. red/blue diverging colormap if you want to highlight positive/negative values.)
- 160-170 is hard to follow.
- Line 256 – "Both methods capture the curvature of the northern side …" Are you referring to the trough?
- Figure 12/13/14 - Not easy to distinguish between red and orange dots. Perhaps plot the mean with a line? And maybe distinguish the years with different symbols. There's too many dots, and it's difficult to compare the years.
- Line 422 - " … other studies that have not shown ASMA bimodality" I suggest revising this statement as Siu and Bowman (2020) have investigated the splitting behavior.

**References**

Manney et al. (2021) A Moments View of Climatology and Variability of the Asian Summer Monsoon Anticyclone

Siu and Bowman (2020) Unsteady Vortex Behavior in the Asian Monsoon Anticyclone

Qie et al. (2025) Weakened Asian summer monsoon anticyclone related to increased anthropogenic aerosol emissions in recent decades

---

## Author Response (AR1)

**Author Comment to Referee #1**

Egusphere-2025-1670, 'Interannual variability of the Asian summer monsoon anticyclone' by O. Kachula et al.

We thank Referee #1 for the positive review and for further guidance on how to revise our manuscript. Our reply to the reviewer comments is listed in detail below. Questions and comments of the referee are shown in bold face.

- 1. Q: The authors present the ASMA interannual variability and properties based on an ERA5 climatology of the time period 1980-2023. They introduce a novel technique to derive the edge of the ASMA region, which is applicable on all time scale and was used in the past to described the polar vortex.
  - A: Many thanks for this summary, but it seem that there is a misunderstanding, our method was *not* used to describe the polar vortex. The technique in the present work is based on Matthewman et al. (2009), but our study is about the Asian Monsoon anticyclone. The reviewer is correct insofar as Matthewman et al. (2009) used absolute vortex moments and applied them to describe the polar vortex.
- 2. Q: Currently, I am wondering why this method should give a threshold that serves the purposes described in L108, i.e. outlining the strongest confinement of the anti-cyclone. How do we know, that this novel analysis is better than the analyses presented before? At various instance it is written that these previous methods produce "noise". Connected to this how do we know, that the ASMA area actually decreases as stated in the abstract?
  - A: We agree that this is an important issue. Our method was compared with two previous studies the PV-based method presented by Ploeger et al. (2015) and the method based on correlations between the Montgomery streamfunction (MSF) and wind speeds presented by Santee et al. (2017). In contrast to Manney et al. (2021) we don't fix a single background (threshold) value for the MSF per vertical level, which Manney et al. (2021) took from Santee et al. (2017) work. To demonstrate the differences between the approach used in Manney et al. (2021) and our approach, we also calculated the background values for each day using the Santee et al. (2017) methodology and compared it with our method as shown in Fig. 1 of this reply.

Choosing then a specific background value for the both methods for one day from the formation phase of the ASMA (in this time period our method produces values higher than Santee et al. (2017)) and subtract it from the MSF grid values, our method preserves less noise compared to the background value calculated using the Santee et al. (2017) methodology as shown in Fig. 2. In our method, the residual MSF values are limited on the Asian monsoon anticyclone, in contrast to Santee et al. (2017) where also positive values for the residual MSF values are present throughout the tropics.

If we choose the background values from the peak phase of the ASMA (our method produces residual MSF values lower than Santee et al. (2017)) our method preserves more information inside of the ASMA box (Fig. 3).

Figure 1: Comparison of the background values  $\mu_b$  using our approach (blue line) and according to Santee et al. (2017) (green dashed line) at 350, 370, 390 and 410 K for the Asian monsoon season 2023.

As a conclusion – we cannot take a single background value because this leads to no information inside of the ASMA box on some days, especially during the formation phase of the ASMA. At the same time using the Santee et al. (2017) methodology to calculate multiple background values leads to some "noise" or loss of information.

To avoid any misunderstandings we revised page 4 lines 108-110 as follows:

The analysis of the interannual variability of the Asian summer monsoon anticyclone depends on the choice of enclosing boundaries of the ASMA. The method proposed here to define the boundaries of the ASMA is based on the absolute vortex moment method that was already used to define the polar vortex.

3. Q: Furthermore, does the optimization method really work to serve the purpose of finding a physically based boundary or does it simply (most of the time) lead to selecting the MSF threshold value that is the maximum value outside of the ASMA box. Then, to me it is not clear why this value should be connected to the strongest circulation.

A: We agree that is an important point. We cannot just simply take the maximum value outside of the ASMA box because there exist strong circulation outside of the ASMA box which, if taken as a background value, would eliminate (or drastically decrease) any information inside of the ASMA box.

4. Q: Is the detailed information on the campaigns needed? Currently, there is a lack of motivation for this and also for showing the 3 [ist seems that some text is missing here]

Figure 2: Comparison of residual MSF values between this work (top) and Santee et al. (2017) (bottom) during the formation phase of the ASMA.

A: We agree that motivation is needed. To clarify this point, we added the following paragraph to Section 3.3:

Three years are highlighted to show that it is possible to provide a comprehensive analysis on a finer time-scale without the need to fix one specific constant value of the MSF. This might help the scientific community to thoroughly analyse the results of those campaigns in future work. This is also the motivation why we need a method that works on any time-scale without depending on averaging the data.

**5. Q: It is stressed, that the novel method works on any time scale. So, do the methods in Manney and Santee as well, correct?**

A: Manney et al. (2021) works with constant climatological background values that might not enclose the boundary during some periods in time as was stated above. The Santee et al. (2017) methodology can be applied to any time scale but as was proved above (Figs. 1, 2 and 3) it preserves "noise" or reduces the information inside of the ASMA box.

**6. Q: This is a personal view: you write about background values (maybe was this the term used before?). I would suggest to call it a threshold value. But at least make clear why the term "background"**

A: We agree both terms are possible, but to be consistent with Matthewman et al. (2009) we decided to preserve their terminology and use the term "background value". We added to the text of the paper:

After calculating MSF background values (the terminology used in Matthewman et al. (2009) to describe the threshold), see page 2.

Figure 3: Comparison of residual MSF values between this work (top) and Santee et al. (2017) (bottom) during the peak phase of the ASMA.

7. The title could be updated to directly mention that an new/updated method is introduced here. This seems to be one of the core information in this manuscript but this is not reflected in the title.

A: Thank you for this suggestion. The new title reads now:

"An optimization-based approach to track the Asian summer monsoon anticyclone across daily and interannual variability"

8. Next to the climatology over the time period 1980 to 2023, three specific years are picked for comparison. The three years are motivated with the aircraft measurement campaigns that took place in the years. However, no data and no conclusions from the campaigns are used for instance to validate the novel method. This raises the question, if the focus on these three years is necessary. It leads to unfulfilled expectations. If these years are given to be able to reference this paper in potential upcoming papers from these missions, then this should be stated.

A: Please, see question 4.

9. Q: In figure 10, the presence of bimodality of the ASMA is shown per season. The bimodality is something shown in observations and therefore should be found in the analysis of the ASMA? There is no statement about whether this is more reasonable than former methods.

A: In addition to per season analysis in our method the days when two peaks are present

in the ASMA are counted, proving that the ASMA often exists as two co-existing clusters rather than one oscillating back and forth that indeed might be seen on per season figures, which was shown as an introduction to the investigation of possible bimodality of ASMA.

10. Q: Absolute Vortex momentum method by Matthewman et al. (2009): Is it applicable for the ASMA in the first place? What are the differences in the polar vortex and the ASMA?

A: It is not directly applicable to the ASMA because the way how the background value is chosen for the polar vortex fails to be useful for the ASMA (also see the answer to your comment 27).

11. Q: Is it planned to provide the data used for this work along with the paper?

A: Yes, the data and calculated background values will be provided. The data will be easily accessible via web page.

12. L5-6: Remove "A 44-year ... by ECMWF." And add info on ERA5 from ECMWF to the next sentence?

A: Done.

13. Q: L14: In my opinion traditionally "two modes" are not the same as "two centres" or "two anticyclones"; the method by Zhang et al. does not allow for multiple centres.

A: Yes, we agree "two modes" and "two centers" are not the same. However, both a western and eastern mode as well as two centers are found in our analysis, in agreement with other previous studies (e.g., Vogel et al., 2015; Ploeger et al., 2015).

14. Eq. 1: Could be introduced in the methodology instead of the introduction? This part feels a little jumpy

A: Done.

15. L20-21: Add references to this sentence.

A: Done.

- 16. L25-28: Please link to the previous sentence to make clear why it is "important" A: Done.
- 17. Q: L40: tautology: defining the edge influences the edge of the anticyclone...; I think the question is rather that you would like to find the boundary of maximum confinement and that there is a debate on how to do that in the best way

A: We agree that the formulation should be changed. But the boundary can still influence the further analysis because the method also provides the residual MSF values encapsulated inside the boundary and hence the position of the edge can impact the center of the mass of the residuals (and the rest of the moments). We changed lines 40-43 accordingly:

There is a debate how to define the edge of the anticyclone best (e.g., Ploeger et al. 2015). The definition impacts the moment quantities in the following analysis and the conclusions that could be drawn on its interannual variability.

**18. Q: L53: "background" value sounds strange "threshold value"**

A: The question was answered above (see point 6). We want to use a formulation consistent with Matthewman et al. (2009).

19. **Q: L56:** do they use different values for different monsoon seasons, please clarify A: No, Manney et al. (2021) use the same constant climatological value for all years and seasons per altitude.

**20. Q: L57: to not enclose the boundaries: do you mean that there is no closed contour?**

A: That means that residual values (Eq. 6) inside of the ASMA box might be zero or near zero. The revised sentence reads now:

The main challenge is to describe the ASMA during development and break-up phases, where MSF background values might be too small to highlight the anticyclone or too large to provide the boundaries at all e.g., residual MSF values inside of the ASMA box might be zero or near zero.

**21. Q: L104: what is daily data at noon? Could the results be influenced by this choice instead of taking full daily averages or performing the analysis on more timesteps per day?**

A: We tried to avoid any averaging routines because we wanted to provide a useful tool to study the ASMA during campaigns which might occur on short time-scales. We did our analysis for noon time every 24 hours, but it is also possible to do our analysis for other time points or use more frequent time steps (e.g., every 6 hours). We added to this sentence:

For our analysis we use ECMWF data at noon (12:00 UTC) but more frequent time-steps (say every 6 hours) are also possible. By using single points in time we tried to avoid any averaging routines because we wanted to provide a useful tool to study the ASMA during short time-scale campaigns.

**22. L106: First two sentences sound very much like in the introduction. Can be removed?**

A: We updated the sentences accordingly.

23. Q: Eq. 2: Is there any area weighting included? Following the description this does not seem to be since delta x is in degree instead of km. Not performing any area weighting while calculating half hemispheric or 3/4 global means seems sounds wrong and potentially affects the results! There is some motivation necessary why the original method of Matthewman is applicable for the ASMA in the first place.

We applied Lambert's azimuthal equal-area projection and compared the data with and without correction. Figure 4 shows MSF residuals at 390K for 06.09.2008. It is evident that the boundary did not change much, although the values near 90°E were weighted down. We also provide comparison table for moments quantities of the same data in Table 1.

At the same time after applying area weighting we see a difference in ASMA duration analysis where the end dates are now in late August instead of September. For detailed discussion, please see 55.

The centroid latitude position of the ASMA changed only for 0.7°, the centroid longitude shows that correction shifts the position further to the east with the difference equal to 5.6°. The change of the excess kurtosis is due to the gap near 90°E after applying the correction. The angle and aspect ration are not affected by the correction much.

We recalculated all results using the area weighting technique and mentioned it in the methodology section:

We use a regular grid  $1^{\circ} \times 1^{\circ}$  resolution. To account for the variation in grid-cell area with latitude, we applied area weighting derived from Lambert Cylindrical Equal-Area projection in our regional calculations.

Figure 4: Comparison of the residual MSF values with (bottom figure) and without (top figure) the area weighting.

**24. Q: L117: integers k and l are not introduced appropriately. Integers "depending on what we want to calculate" – please elaborate further**

A: In our opinion the purpose of k and l is self explanatory from Eq. 2. These two integers are just a way to compactly write one equation instead of multiple and this notation was taken from Matthewman et al. (2009).

Consider the case when k = 0 and l = 0:

|                    | Without correction | With Correction |
|--------------------|--------------------|-----------------|
| Centroid Latitude  | 30.6°              | 29.9°           |
| Centroid Longitude | 74.8°              | 69.2°           |
| Excess kurtosis    | -0.08              | 0.15            |
| Angle              | $-0.06^{\circ}$    | -0.06°          |
| Aspect ratio       | 8.97               | 8.92            |

Table 1: Comparison of the moments quantities after applying the area weighting correction.

$$M_{00} = \sum_{i=0}^{n-1} \sum_{j=0}^{m-1} (\hat{\mu}_{i,j} - \mu_b) \Delta y \Delta x, \tag{1}$$

 $y_i^k$  and  $x_j^l$  vectors are equal to 1 so we can omit them from the equation. What is left is just a double sum over residuals. If we want to calculate a weighted arithmetic mean of the coordinate we "switch on" k or l e.g.,

$$M_{10} = \sum_{i=0}^{n-1} \sum_{j=0}^{m-1} (\hat{\mu}_{i,j} - \mu_b) y_i \Delta y \Delta x,$$
 (2)

or

$$M_{01} = \sum_{i=0}^{n-1} \sum_{j=0}^{m-1} (\hat{\mu}_{i,j} - \mu_b) x_j \Delta y \Delta x.$$
 (3)

See updated page 5.

**25. Q: L125 What is meant with "bipolarity".**

A: In this context it means the splitting-like behavior of the two centers.

Text changed in response:

The excess kurtosis (EK) serves as a measure of how far the shape is from the ellipse and can be used to investigate the bipolarity (in other words the splitting like behavior of the centroid) of the MSF distribution, ASMA splitting behavior into two anticyclones or strong eddy shedding events

**26. Q: L126: What are eddy shedding events? They are not mentioned in the results, but in the section title.**

A: Eddy shedding events are now mentioned in the introduction and (in response) we added here references (e.g., Popovic and Plumb (2001); Vogel et al. (2014); Riese et al. (2025)).

**27. Q: L132-137: I do not understand what these lines are supposed to tell the reader.**

A: This lines answers the question (see 10) why Matthewman et al. (2009) method is not applicable to the ASMA in the first place.

**28. Q: L142: Necessary to define the ASMA region over the quarter of the globe? 60/65°N would be enough? Following Fig. 6, the region should reach eastward of 0° E.**

A: In our opinion it was interesting to encapsulate as much information inside of the ASMA box as possible. At the same time the sensitivity analysis illustrated on Fig. B1 shows that shifting the box southward (hence removing 65°N-90°N part) does not affect the optimized background value.

**29. Q: L145: Don't understand what k and l mean. It is M\_in and M\_out**

A: Please see point 24.

**30. Q: L146: So, the idea means, that the other regions of the globe with larger streamfunctions are weighted down by the large area (3/4 of the globe) with low Montgomery streamfunction?**

A: There seem to be a misunderstanding here. Two cases were described in the paper. Case one is when MSF values inside of the ASMA box are higher than values outside of the box. Case 2 is when there is a cluster of high values outside of the ASMA box. In both cases the method tries to find such a background value that maximizes the objective function. The value for MSF originating from this optimization can then enclose the ASMA and in addition (if present) the cluster outside of the box.

**31. Q: Caption Fig. 1: The statement "There MSF values inside of the box are larger than outside." is wrong as this statement is too simplified.**

A: Consider MSF value for 25.06.2022 at 390 K. Table 2 (of this reply) shows max, min and mean values inside and outside of the box.

|      | MSF inside of the ASMA box, $10^3 \mathrm{m}^2/\mathrm{s}^2$ | MSF outside of the ASMA box, $10^3 \mathrm{m}^2/\mathrm{s}^2$ |
|------|--------------------------------------------------------------|---------------------------------------------------------------|
| Max  | 368.3                                                        | 366.7                                                         |
| Min  | 360.5                                                        | 346.1                                                         |
| Mean | 364.6                                                        | 359.8                                                         |

Table 2: Values of the MSF grid data for 25.06.2022 at 390 K.

Obviously, there are values inside of the ASMA box that are higher than values outside of the box. Thus we don't see that the last sentence in the caption is incorrect.

**32. L152: Introduce/cite Dual Annealing algorithm**

A: Done.

**33. Q: L154: time dependence of Eq. 1 not mentioned before. Add it or mention it**

A: Are you referring to Eq. 2? There is no time dependence. L154 just states that we work with the same MSF grid values during the optimization step. We changed text in response on page 6.

**34. Q: L164f: I can hardly imagine, that (almost) all MSF outside of the ASMA box are zero, and none inside is. The sum is maxed to differ the most. Not necessary for such an absolute statement**

A: We provide an animation (Animation1.gif in the attachments) to prove our point. The illustrated example shows how subtracting different background values (slowly increasing and starting from  $340 \times 10^3 \,\mathrm{m}^2/\mathrm{s}^2$ ) affects the residual values. As can be seen the residual values of the MSF outside of the ASMA box during the optimization process converge to zero faster than the residual values inside of the ASMA box.

35. Q: L174: "small noise": Is this really small noise? Isn't this how the optimization is supposed to work. Otherwise simply the maximum value outside the ASMA box could be used as threshold value. (see general comment)

A: This means that optimized background value sometimes preserve non zero residual values outside of the box e.g., within the tropics. Also, see points 2 and 3.

The sentence was changed to:

Sometimes even  $\widetilde{\mu}_b$  preserves small non-zero residual values (i.e., noise) outside of the box or another circulation is also present as was noted before

**36. L180 onwards strange?**

Please, clarify.

37. Q: L196 (and following): "unwanted noise" in Santee et al. 2017 How can you be sure, that this is true? Why can you be sure that your method is really better.

A: The question was answered above. See point 2

38. Q: Fig. 3: Why only Santee and not (also) Manney? This is the only time Santee is used as comparison. Why dashed line? To account for colour-blindness?

A: Manney et al. (2021) work uses the background values provided by Santee et al. (2017). Yes, the dashed line is used to account for colour-blindness. We added to the sentence:

(note that Manney et al. (2021) use the background value  $\mu_b$  provided by Santee et al. (2017))

39. Q: L203f: Show results of Ploeger et al. (2015) in Fig. 3 as well?

A: The results of Ploeger et al. (2015) cannot be shown in Fig. 3 because this method is based on PV and not on Montgomery streamfunction. We only can qualitatively compare the boundaries of both methods which are shown in Fig. 8 and C2.

40. Q: L213: What does robust mean in this case?

A: The area time-series in our work are different compared to Manney et al. (2021) and following their methodology requires much more assumptions and preprocessing and still led to wrong results where start dates were in September or end dates were in May for some years. Using our methodology to find start/end dates we achieved smaller standard deviation and for majority of years the starting dates end up in May-June range and end dates – in late August-September range. This is what robust means here.

41. Q: L213: What index? The anticyclone area? Clarify. Can you explain why robust start end days could not be identified via the area? Also, the PCA is not described in detail and it would be good to know why 14 days are set.

A: The index here means the first principal component that we get after applying PCA. The impossibility to get start/end dates from the area time-series is explained above. PCA is

widely used technique and its description can be found in specialized literature. We tried different number of consecutive days and found that 14 days yields results that are similar to Manney et al. (2021).

**42. Q: L215: Is it reasonable to average pressure and PV over 1/4 of the globe?**

A: To retrieve the index we tried different sets of parameters and found that the described set works better in our case.

**43. Q: L217: larger than zero?**

A: We want to find the time points when the data goes above and below horizontal line at zero for 14 consecutive days.

**44. Q: L217: Why 14 days? Did you test other time periods? How sensitive is your results on this threshold?**

A: Yes, we tried different number of days. Unfortunately, even after PCA the data has high variability and it indeed is sensitive to smaller number of days. Two weeks give results similar to Manney et al. (2021).

**45. Q: L238f: Shift is present in Northward es well as Eastward direction. Why only northward shift mentioned?**

A: We agree, we should be more accurate here. We revised this sentence as follows.

A northward shift of the ASAM center is found at the beginning of the Asian summer monsoon and a southward shift is found at the end of the monsoon season in agreement with previous studies (e.g., Vogel et al., 2015; Goswami, 2012, and references therein)

**46. Q: L258: "does not capture the western tail of the ASMA". This is present in this work, but not certain, that is should be there. Rephrase it, so it is more objective. You find a difference.**

A: We changed the text:

In contrast to Ploeger et al. (2015) our method suggest a western tail of the ASMA between  $0^{\circ}E - 10^{\circ}E$ .

**47. Q: Fig. 9: Caption. Reddish colour is probably not the centroid longitudinal position, but the cluster of the maximum residual MSF. Otherwise this does not make sense for bimodal case. Would agree to L285**

We agree with this comment. The revised caption now reads:

Reddish colors indicate where the maximum residual MSF values tend to cluster.

**48. Q: Fig. 10: To optimized background values are shown, but not/ helpful to understand the left plot. Nothing necessary to see directly. Put into appendix?**

A: We agree with this comment and have moved the subplots to the appendix.

**49. Q: L284: The threshold is set arbitrarily?**

A: We agree in principle; in this case finding an objective threshold is not possible. We explain better now:

(50% of the max value as a reasonable threshold to check where the residual MSF values at least as half of its maximum are placed)

**50. Q: L327: "despite their higher variability". Is this not always true?**

A: In general individual years always have high standard deviation, so this part of the sentence was omitted, in response:

The highlighted years (2017, 2022, 2023) generally tend to group around the climatology.

**51. Q: L343ff: I do not understand the message of the sentence**

A: During the formation and break-up phases the excess kurtosis cannot be a reliable description of the ASMA. The revised text:

During the developing phase of the ASMA up to until June and also during the break up phase after September the standard deviation of EK (and thus its variability) is higher than in the ASMA peak phase during the June-September period at all vertical levels and cannot be viewed as a reliable description of the ASMA evolution.

**52. Q: L354-L359: This part can be shortened. Good agreement with Manney et al 2021.**

A: We agree. We changed this part with the following text:

Figures 12, 13 and 14 show a good overall agreement with the results of the study by Manney et al. (2021). Our method gives us an indication of the ASMA's evolution between May and September despite a relatively strong variability. Both centroid latitude and longitude and the aspect ratio show similar temporal evolution as the Manney et al. (2021) findings. Although the area of the ASMA occupies around 10% of the hemisphere in the beginning of August in our results and Manney et al. (2021) (MERRA-2), the shape of the time-series is more flat in this work during the peak phase of the ASMA compared to Manney et al. (2021).

**53. Q: L362ff: Generally true, but the focus on the years does not suit the overall results**

A: In response we changed the text accordingly:

Because we provide individual campaign years in addition to climatological time-series, our results show oscillations of excess kurtosis and angle quantities compared to Manney et al. (2021), who focus on the climatological feature of the ASMA behavior.

**54. Q: L369: What is the index? "the/a" PCA technique. Not well enough introduced in Sec. 2**

A: This was explained above. Please see point 41. In response we clarified "the index":

The time-series of first principal component (the index) was calculated using PCA technique.

**55. Q: L379: Did you test for statistical significance?**

A: We performed permutation analysis for start, end dates and duration period at 370, 390 and 410 K (Fig. 5 of this reply). For each case a set of randomly permuted time-series was created with 100 000 elements. Then we calculated the slope for each time-series in a set and built a histogram that shows the distribution of slopes. The red vertical line denotes the slope of the original time-series. We also calculated two-sided p-values that can be seen on each subfigure.

Figure 5 shows that only 370 K has a low p-value. The slopes of the time-series at 370 K for end dates and duration show relative strong trend and hence are statistical significant. The

time-series at 390 K and 410 K in contrast has small variability between years, and hence the high p-values tell us that further analysis is needed to make any conclusions about the duration of the ASMA.

We added the following text to the section:

We performed permutation test on the ASMA start, end dates and the duration at 370, 390 and 410 K to assess statistical significance. Only time-series at 370 K was statistical significant (p < 0.05). Time-series at the higher altitudes (390 K and 410 K) showed limited interannual variability and weak trends, indicating that additional analysis is needed before drawing any conclusions about ASMA duration at those levels.

At the same time the calculation of the ASMA duration is not robust. It is sensitive to initial conditions which also can be seen in Manney et al. (2020) work where slopes of the start, end and duration data vary with chosen reanalysis. We added the following text to the section:

Our calculations (Fig. 14) show similar results at 370 K where we found largest trends (we do not provide the data for 350 K because this level is below the main anticyclone). There are mixed results at higher altitudes where end dates show negative trend at 390 K and positive trend at 410 K. Overall, the duration is sensitive to initial conditions (e.g., the quantity timeseries or the threshold). Manney et al. (2020) show that the slopes of the duration also vary respect to the reanalysis (MERRA-2, ERA-I or JRA-55).

**56. Q: Fig. 16: Neutral ENSO case is for exactly for value 0.5? So many cases are neutral?**

A: There is a typo in the caption, La Niña years are considered when the temperature is less than  $-0.5^{\circ}$ C, so the neutral years are in the range [-0.5, 0.5]. The typo is corrected. The new caption reads now:

Trends of the ASMA area, centroid latitude and longitude for JJA based on ERA5 reanalysis. Color marks ENSO type of a year based on ONI index (using DJF, from the winter before the considered monsoon period). The year is marked as El Niño if the index is  $> 0.5^{\circ}C$ , La Niña if the index is  $

Figure 5: Permutation analysis of the slopes of the start, end dates and the duration of the ASMA at 370, 390 and 410 K. The red line denotes the slope of the original time-series. Two-sided p-value is showed in the upper-right corner of each subfigure.

**Author Comment to Referee #2**

Egusphere-2025-1670, 'Interannual variability of the Asian summer monsoon anticyclone' by O. Kachula et al.

We thank Referee #2 for the positive review and for further guidance on how to revise our manuscript. Our reply to the reviewer comments is listed in detail below. Questions and comments of the referee are shown in bold face.

1. Siu and Bowman (2020) investigates the modality of ASMA and has some extensive discussions on the variability of ASMA having one, two, or more vortices. I notice that you do not refer to this work; I suggest looking at their research and adding some discussion on consistencies or differences between your work and theirs.

A: We would like to thank reviewer #2 for drawing our attention to this paper. We added the discussion about the modality of ASMA investigated in this work to the Discussion section:

Siu and Bowman (2020) developed an algorithm that is able to track multiple simultaneous subvortices of the ASMA. They used the ERA-Interim reanalysis and showed that the ASMA contains two or three distinct subvortices 69% of the time simultaneously. Siu and Bowman (2020) show that in many cases there is two broad peaks of the ASMA – between about 40° E and 100° E and near 150° E.

Our results are similar to Siu and Bowman (2020) but instead we work with residual MSF on isentropic surface and consider first climatological horizontal plots to see qualitatively the existence of multiple peaks at 370, 390 and 410 K between June and September (Fig. 4; Fig. C1); the Hovmöller diagram of normalized zonal mean residual MSF values for each year to see the distribution of peaks and count number of days when there is a simultaneous presence of two peaks on the both sides relative to the longitudinal centroid separation line. It is clear that there is clustering of peaks during 1980-2023 near 50°E and near 150°E (for some years) that indicates a bimodal state of the ASMA similar to the findings Siu and Bowman (2020).

2. In addition a recent work Qie et al. (2025) also investigated the trend of the ASMA. I think incorporated the finding of Qie et al. in their discussion and result interpretation.

A: Thank you for pointing out this work. Unfortunately, it is hard to directly compare Qie et al. (2025) and our results due to differences in methodologies. Qie et al. (2025) consider geopotential height anomalies, PV and stream function and work with zonal mean values to show that there is a significant weakening of the ASMA. In our study we show the temporal and spatial evolution of the ASMA and the only thing that can potentially align with Qie et al. (2025) study is the fact that the ASMA area has decreased in size over the period 1980-2023.

3. Throughout the interpretation of results, the three flight campaigns (StratoClim, ACCLIP, PHILEAS) are mentioned and the corresponding years are chosen for analysis, but your results for these years lack impact if we don't know what the

campaign had found. Are any of the campaign's results explainable with your findings? If so I think they should be included in your discussions.

A: We showed the individual years during with the campaigns took place only to illustrate that our method can be used for analysis on a finer time-scale without requiring averaging procedures. To clarify this point, we added the following paragraph to Section 3.3:

Three years are highlighted when measurement campaigns took place to show that it is possible to provide a comprehensive analysis on a finer time-scale without the need to fix one specific constant value of the MSF. This might help the scientific community to thoroughly analyse the results of those campaigns in future work. This is also the motivation why a method is needed that works on any time-scale without depending on averaging the data.

A strong variability are found in the centroid position of the ASMA latitude and longitude for the individual years 2017, 2022 and 2023 (Fig. 12) with an increased variability of the latter two years. In all three years time periods exist in which the variability of the centroid position of the ASMA is lower or greater than the standard deviation of the climatological data.

In particular during August 2022, there is a strong shift of the ASMA to the east compared to the climatological mean data. Therefore aircraft flights inside the ASMA could be performed during the ACCLIP campaign over South Korea (Pan et. al, 2024) contrary to the original campaign goal to measure the eastern outflow of the ASMA.

But also during the PHILEAS campaign (Riese et al. 2025) to measure the eastern outflow of the ASAM over Alaska, there was a strong shift of the ASMA to the east compared to the climatological mean data during the first half of both August and September 2023.

4. It seems like one of the advantages of your method is its ability to capture complex shapes of the ASMA. For instance, in Figure 8 your method shows the tail to the west at 30N while the Ploeger method does not.

A: Yes, we updated the text to address this finding:

In contrast to Ploeger et al. (2015) our method suggests a western tail of the ASMA between  $0^{\circ}E - 10^{\circ}E$ .

5. Line 142: ASIA box is from 0E to 180E. Though this is not the norm but we often observe anticyclone stretching west of 0E. You can see this in your Figure 6. Is it possible to shift the box westward?

A: There are different ways to define the ASMA box. For example, Santee et al. (2017) define ASMA region  $(15^{\circ} - 45^{\circ}N, 10^{\circ} - 130^{\circ}E)$ , Nützel et al.  $(2016) - (15^{\circ} - 45^{\circ}N, 30^{\circ} - 140^{\circ}E)$ . In this work a larger ASMA box was used  $(0^{\circ} - 90^{\circ}N, 0^{\circ} - 180^{\circ}E)$  but in addition we provide the sensitivity analysis (Appendix B) to show the limits of the method by moving the ASMA box position. It is possible to shift the box westward but in our opinion it won't affect the determination of the optimized background value.

6. Starting at line 196, the statement "During the developing (May-June)... the Santee et al. (2017) method gives lower values than our method, which means that some unwanted noise is still preserved... "Though Fig 3 does show lower values the Santee method, I think it should be explicitly shown (perhaps with some cases) that the method Santee et. al indeed performs worse. Likewise, for

the anticyclone peak phase, the authors can show that their method defines the anticyclone better.

A: We can provide two examples: in the first case we choose the background values for both methods during the formation phase of the ASMA. Figure 3 from the paper shows that the optimized background value is higher than Santee et al. (2017); in the second case we choose the background values during the peak phase (the optimized background value is lower than Santee et al. (2017)). Fig. 1 (top) of this reply shows the residual MSF values for the first case. The Santee et al. (2017) methodology (bottom) preserves "small noise" compared to this work. Fig. 2 of this reply shows the second case in which this work outlines a larger area of the ASMA than for the Santee et al. (2017) methodology.

Figure 1: Comparison of residual MSF values between this work (top) and Santee et al. (2017) (bottom) during the formation phase of the ASMA.

The explanation was added to the appendix of the paper.

7. Line 202 states that the ASMA at 350 K does not exist. I think this statement is not true. Figure 1 below is Montgomery streamfunction at 350 K and it is evident that a closed contour (i.e. anticyclone exists).

A: We agree that the formulation should be changed. The ASMA at 350 K is weak and hard to capture for every day. The difficulty of working with the ASMA at 350 K can be even seen in Fig. 3 of the paper. There is small variability of the optimized background value during the whole season. We updated the sentence accordingly:

Our method shows  $\widetilde{\mu}_b$  values for 350 K that have a small variation during the whole Asian monsoon season reflecting that at 350 K the ASMA is weak and capturing it poses a challenge.

Figure 2: Comparison of residual MSF values between this work (top) and Santee et al. (2017) (bottom) during the peak phase of the ASMA.

8. Line 250; Figure 5/6/7 - The "angle" quantity is discussed without much context. I think there needs to be some explanation on how to interpret this angle (perhaps a schematic or idealized cases) and what its implications are. For Figure 6, why is the angle related to the curvature at 60E but not the western side (e.g. 40E?).

A: Matthewman et al. (2009) define  $\psi$  as the angle between the x axis and the major axis of the ellipse which can be obtained as

$$\psi = \frac{1}{2} \tan^{-1} \left( \frac{2J_{11}}{J_{20} - J_{02}} \right),$$

where  $J_{kl}$  is the relative vortex moments:

$$J_{kl} = \int \int [\hat{q}(x,y) - q_b](x - \bar{x})^k (y - \bar{y})^l dx dy,$$

where q is the PV field used in Matthewman et al. (2009).

As you can see, the residuals  $(\hat{\mu}(x,y) - \mu_b)$  are involved in the calculations, which means that not the contour per se impacts the angle but the distribution of weights inside of the contour.

The revised introduction of the angle (Line 126) reads now:

"the angle between the equatorial axis and the major axis of the ellipse (the calculation of which depends not only on the boundary of the ASMA but also on the residual MSF values. See A2 and A3)"

9. Figure 9/10 – Could you consider adding a subplot to show whether a JJA has 1, 2, or more maxima? I wouldn't demand this be done but think it can help us see whether a specific JJA tended to be bimodal or not.

A: Figures 9 and 10 do show the number of maxima and their position for each year. The coefficient is given in fraction relative to an individual year's max value. For the discussion we can plot zonal mean residual MSF for 1994 at 370 K.

Figure 3: Zonal mean  $(15^{\circ}N - 45^{\circ}N)$  residual MSF values in JJA 1994 at 370 K.

On Fig. 3 of this reply we can clearly see that during JJA months the anticyclone's residual MSF values clustered near 3 positions -  $75^{\circ}E$ ,  $95^{\circ}E$  and  $140^{\circ}E$ . This fact doesn't yet prove the bimodality of the anticyclone because the ASMA might have just shifted during JJA along longitudinal axis. For this reason we provide Fig. 11 in the paper that counts whether a specific day in a year is bimodal or not.

10. Line 346 – "The temporal evolution of the EK during 2022 shows that in June-September period there are more splitting like events than in 2023 at 390K." Do you infer this by seeing more higher values of blue dots verses red dots? Just asking for clarification.

A: Yes, especially at 390 K it is clear that for 2022 (blue dots) there is more peaks during June-September than for 2023 (red dots).

11. There are many short paragraphs, some with one sentence. I suggest revising the writing to reduce the number of these.

A: Thank you for your suggestion. We adjusted the text accordingly.

12. Line 102 – Is this supposed to be 0.25 degrees?

A: Thank you for the correction. Indeed, the horizontal resolution should be  $0.25^{\circ} \times 0.25^{\circ}$ .

13. Figure 5, 6: I suggest changing the colormap. The choice of colormap depends on what you want to highlight (i.e. red/blue diverging colormap if you want to highlight positive/negative values.)

A: Majority of the PV values in the region are positive or near zero so in our opinion the change of the colormap won't provide additional details.

**14. **160-170** is hard to follow.**

A: Thank you for your feedback. We changed the paragraph with the following text:

"We discuss two cases of MSF grid data:

- 1. Many of the grid values inside of the ASMA box are larger than values outside of the ASMA box.
- 2. Regions outside of the ASMA box have MSF values that are comparable or larger to the values inside of the ASMA box.

Let's consider two specific days that correspond to the above described scenario: 1st August 2008 at 390 K (case 1) and 6th September 2008 at 390 K (case 2). We can sample the objective function for the given days – Fig. 1 and Fig. 2 respectively.

For case 1 the objective function is low for small  $\mu_b$  which means that the sum of all values inside of the ASMA box are comparable to the sum of all values outside of the ASMA box. But eventually, when we increase  $\mu_b$  (e.g., subtracting higher value from the given MSF grid data) the values outside of the ASMA box converge to zero more rapidly than the values inside of the ASMA box and the objective function rise (Fig. 1). At some point we reach such  $\mu_b$  that will give us the highest sum of residual MSF values inside of the ASMA box while keeping the values outside of the ASMA box near zero. This  $\mu_b$  value gives us a position of the maximum of the objective function. Increasing further  $\mu_b$  will just decrease the values inside of the ASMA box (the values outside of the ASMA box are already zero) so the objective function starts to decrease and reaches zero.

For case 2, when a strong circulation outside of the ASMA box is present, the same steps described for case 1 remain true with the exception that when we reach some relative high  $\mu_b$  the values outside of the ASMA box do not become zero. This is why the objective function has a different shape but still the global maximum points at such  $\mu_b$  that will keep the sum of all residual MSF values inside of the box highest while the sum of residual MSF values outside of the ASMA box will be the lowest.

Both cases allows us to find the optimized background value  $\tilde{\mu}_b$  that will provide a contour line that will encircle the ASMA."

**15. Line 256 – "Both methods capture the curvature of the northern side ..." Are you referring to the trough?**

A: The sentence tries to point out that both methods shows the push of the northern part of the boundary near  $80^{\circ}E$  inside of the anticyclone and how both methods follow the wind velocity vectors there.

16. Figure 12/13/14 - Not easy to distinguish between red and orange dots. Perhaps plot the mean with a line? And maybe distinguish the years with different symbols. There's too many dots, and it's difficult to compare the years.

A: Many thanks for your feedback. We changed the Figures 12, 13, 14 so the data can be better distinguished. On Fig. 4 of this reply you can see an example of the new style:

Figure 4: Aspect ratio time-series with new markers.

**17. Line 422 - " ... other studies that have not shown ASMA bimodality" I suggest revising this statement as Siu and Bowman (2020) have investigated the splitting behavior.**

A: Thank you for this suggestion. We updated the sentence accordingly:

We show that horizontal plots of the residual MSF values provide the information about splitting-like behavior of the anticyclone in July, August and September supporting the recent study of Siu and Bowman (2020) who showed multiple simultaneous peaks of the ASMA.

**References**

- Matthewman, N. J., Esler, J. G., Charlton-Perez, A. J., and Polvani, L. M.: A New Look at Stratospheric Sudden Warmings. Part III: Polar Vortex Evolution and Vertical Structure, Journal of Climate, 22, 1566 1585, https://doi.org/10.1175/2008JCLI2365.1, 2009.
- Nützel, M., Dameris, M., and Garny, H.: Movement, drivers and bimodality of the South Asian High, Atmos. Chem. Phys., 16, 14755–14774, https://doi.org/10.5194/acp-16-14755-2016, 2016.
- Ploeger, F., Gottschling, C., Grießbach, S., Grooß, J.-U., Günther, G., Konopka, P., Müller, R., Riese, M., Stroh, F., Tao, M., Ungermann, J., Vogel, B., and von Hobe, M.: A potential vorticity-based determination of the transport barrier in the Asian summer monsoon anticyclone, Atmos. Chem. Phys., 15, 13145–13159, https://doi.org/10.5194/acp-15-13145-2015, 2015.
- Santee, M. L., Manney, G. L., Livesey, N. J., Schwartz, M. J., Neu, J. L., and Read, W. G.: A comprehensive overview of the climatological composition of the Asian summer monsoon anticyclone based on 10 years of Aura Microwave Limb Sounder measurements, J. Geophys. Res., 122, 5491–5514, https://doi.org/10.1002/2016JD026408, 2017.
- Siu, L. W. and Bowman, K. P.: Unsteady Vortex Behavior in the Asian Monsoon Anticyclone, Journal of the Atmospheric Sciences, 77, 4067 4088, https://doi.org/10.1175/JAS-D-19-0349.1, 2020.

**Author Comment to Referee #3**

Egusphere-2025-1670, 'Interannual variability of the Asian summer monsoon anticyclone' by O. Kachula et al.

We thank Referee #3 for the positive review and for further guidance on how to revise our manuscript. Our reply to the reviewer comments is listed in detail below. Questions and comments of the referee are shown in bold face.

1. What is the purpose of defining the ASMA on a day-to-day basis? The main emphasis of the recent work is the interannual variability and long-term trend in the ASMA boundaries, which can be simply obtained by the monthly representation of the ASMA boundaries using the existing methods.

A: Our motivation to find a method that can define the ASMA on any time scale, not only on a day-to-day basis, is to provide a tool that does not depend on averaging the data. The proposed method can be useful for measurement campaigns with a typical duration of several weeks and provides a finer temporal resolution of days or even hours.

2. The title of the paper can be modified as "Temporal variability of the Asian summer monsoon anticyclone" as it covers variation from day to day to long-term trends.

A: We thank reviewer #3 for this proposal and revised the title accordingly. The new title is "An optimization-based approach to track the Asian summer monsoon anticyclone across daily and interannual variability"

3. The present study states that the importance of defining the ASMA boundaries; however, they did not discuss the recent work by Musaid et al. (2024) providing the definition for ASMA boundaries based on the Jet stream cores. Provide the citations sentence in the L42-43: "Different ways to define the boundary of the ASMA exist"

A: We thank reviewer #3 for pointing out to different studies to define ASMA boundaries. Musaid et al. (2024) is cited in the revised version of the manuscript.

4. L 60-64: The Authors proposed a method to identify MSF background values for an individual point in time per isentropic surface, which is an optimised MSF value to describe the ASMA boundaries, which can capture its day-to-day variability. This is one important aspect of Musaid et al. (2024) to study to day variation of the ASMA to understand its variation during active and break phases of the monsoon. They also proposed a new GPH method to define the ASMA for the active and break phases of the monsoon based on the GPH values at the tropical easterly jet (TEJ) and subtropical westerly jet (STJ) locations. Their main conclusion is that other methods are not suitable to capture the ASMA boundaries on a shorter scale, especially on the day-to-day scale. This aspect needs to be discussed here.

A: It is beyond the scope of our work to compare our methodology with all existing methods that can define the ASMA boundaries. We did a comparison for a few selected methods in

particular Santee et al. (2017) and Manney et al. (2021) that also use MSF to demonstrate the advantages of our method. Our intention was to develop a tool, which is simple to use that can encircle the ASMA on any time scale. And our method requires for this only the Montgomery streamfunction that incorporates geopotential, temperature and the specific heat capacity at constant pressure.

5. L83-90: The title of this paper emphasises on Interannual variability of the ASMA; however, the authors fail to provide novelty, rather they end up highlighting the meridional transport of the tracers and pollutants to the global stratosphere. Authors need to focus and provide the main objectives carried out in this paper.

A: L83-90 (preprint version), as well as description of the StratoClim and ACCLIP campaigns above, provide the motivation for developing the method that can be suitable for short time scale measurement campaigns and doesn't require averaging in time. The paper provides the intra-seasonal and interannual analysis based on the suggested methodology and shows that the spatio-temporal evolution of the ASMA aligns with previous studies. Moreover our work supports recent evidence of a bimodality of the ASMA and discusses new results concerning the interannual ASMA area decrease supporting another recent paper of Qie et al. (2025) who show the weakening some trends of ASMA quantities.

6. L104: Why specific time 12:00 UTC, considered for the analysis instead of the daily average?

A: Please see answers 1 and 4 that explain why we avoid data averaging.

7. L230: Figure 4 describes climatological mean residual MSF; however, the description provided is very brief.

A: Figure 4 used as an example of residual MSF values and as an introduction to possible bimodality of the ASMA which is discussed in more details in section 3.2.

8. L245: Traditionally, researchers use the withdrawal phase instead of the break-up phase of the monsoon. So, the sentence can be revised as "During the withdrawal phase in September, the shape of the mean residual MSF is elongated and shrunk along the latitude axis, indicating the break-up of the anticyclonic circulation. It is suggested to use the withdrawal phase instead of the breakup phase throughout the manuscript.

A: Thank you for this suggestion, however both terms can be found in literature. The withdrawal phase related to the monsoon rainfall so we prefer to keep the break-up phase terminology to emphasize the relation to the Asian summer monsoon anticyclone.

9. L248: The authors mentioned that residual MSF (L232) is not used to obtain the ASMA boundaries. How are ASMA boundaries obtained?

A: It seems there is a misunderstanding here. The sentence in L232 is:

The mean residual MSF does not represent the optimized boundaries of the ASMA at any specific time-point and are used here only for a qualitative description of the ASMA development from pre-monsoon to post-monsoon.

Note that the sentence (l. 232, submitted version) is on the *mean* residual MSF value. The sentence clearly states that if we average multiple days of the residual MSF values the result won't represent the ASMA boundary of any specific point in time.

10. Descriptions about Figures 5-7 are very short, they can be kept as subplots in one figure. Figure 8 can also be included. Why not comparisons with existing methods discussed here?

We prefer not to merge Figures 5-7 due to figures scalability issues and potential loss of clarity. Figures 8 and C2 show the comparison with existing methods (e.g. Ploeger et al., 2015) and L254-262 contain the discussion.

11. L250: change 15.07.2017 as 15 July 2017 and elsewhere in the manuscript.

A: We agree, the correct format for dates in ACP publications is '15 July 2017'. We revised the manuscript accordingly.

12. L255: Replace "our boundary" with "ASMA boundary obtained using proposed method"

A: done

13. L261: Replace "our method" with "proposed method"

A: done

14. L255-262: Rewrite these sentences

A: This reviewer's comment is unspecific and unclear. Please clarify.

15. L261: What do you mean by aligns with wind velocity?

A: By aligning with wind velocity we wanted to underline that the curvature of the boundary match with the wind velocity vector field. To be more clear, we revised this sentence: "... and the ASMA boundary deduced here aligns with the wind velocity..."

16. L271 & Figures 9-10: The Authors show the seasonal mean representations of the residual MSF based on the proposed method. How are these representations different from the one obtained from the existing method? It provides a concise way to interannual variations in the ASMA boundaries as well as its horizontal structure, demonstrating the bimodality during different years.

A: Thank you for the suggestion. We revised our manuscript by comparing our findings with Siu and Bowman (2020) work.

17. Figure 11: "Number of days per year during monsoon season (JJA)" would be more meaningful. Do you observe bimodality during May and September when ASMA is very weak?

A: Thank you for this suggestion. Figure 4 shows that May and September month doesn't have strongly pronounced clustering of the residual MSF values near multiple centers.

**18. L319-321: Rewrite the sentences**

A: The sentence was revised as follows: A south-northward shift (and back) as well as eastwest oscillation are found in the centroid position of the ASMA latitude and longitude for the individual years 2017, 2022 and 2023 (Fig. 12). It is known, that also Asian summer monsoon rainfall has a south-northward shift (and back) and east-west oscillations (Goswami 2012).

19. L354: Overall, a good agreement. Delete study; L354-355: "Our method...variability" is redundant; L355-60: The Authors claim the comparison with Manney et al. (2021) needs to be rewritten.

A: We changed this paragraph as follows:

Figures 12, 13 and 14 show a good overall agreement with the results of the study by Manney et al. (2021). Our method gives us an indication of the ASMA's evolution between May and September despite a relatively strong variability. Both centroid latitude and longitude and the aspect ratio show a similar temporal evolution as found by Manney et al. (2021). Although the area of the ASMA occupies around 10% of the hemisphere in the beginning of August in our results and the results of Manney et al. (2021) (based on MERRA-2), the shape of the time-series is more flat in our work during the peak phase of the ASMA compared to Manney et al. (2021).

20. Subsection 3.4: (at 370 K; Fig. 15) What is the significance of the trends in the start date and end dates? Why are start dates becoming early (In May) and end dates late (in September-October) over the year?

A: We agree with the reviewer that the most relevant quantity is the duration of the monsoon. But the duration is of course a combination of start and end dates. The question why start and end dates are changing is not easy to answer; the trend in start and end dates are first of all a product of the present analysis and the statistical verification of the trends (see below). Further, we performed a permutation analysis for start, end dates and duration period at 370, 390 and 410 K (Fig. 1 of this reply). For each case a set of randomly permuted timeseries was created with 100000 elements. Then we calculated the slope for each time-series in a set and built a histogram that shows the distribution of slopes. The red vertical line denotes the slope of the original time-series. We also calculated two-sided p-values that can be seen on each subfigure.

Figure 1 of this reply shows that only 370 K has a low p-value. The slopes of the time-series at 370 K for end dates and duration show relative strong trend and hence are statistical significant. The time-series at 390 K and 410 K in contrast have a small variability between years, and hence the high p-values tells us that further analysis is needed to make any conclusions about the duration of the ASMA at these theta levels.

- 21. Subsection: 3.5 Interannual variability and trends of the ASMA area and location A: Done.
- 22. Section 4: Discussion and Conclusions Provide implications towards the conclusions obtained in the present study

A: Done.

**References**

- Manney, G. L., Santee, M. L., Lawrence, Z. D., Wargan, K., and Schwartz, M. J.: A Moments View of Climatology and Variability of the Asian Summer Monsoon Anticyclone, Journal of Climate, 34, 7821 7841, https://doi.org/10.1175/JCLI-D-20-0729.1, 2021.
- Siu, L. W. and Bowman, K. P.: Unsteady Vortex Behavior in the Asian Monsoon Anticyclone, Journal of the Atmospheric Sciences, 77, 4067 4088, https://doi.org/10.1175/JAS-D-19-0349.1, 2020.

Figure 1: Permutation analysis of the slopes of the start, end dates and the duration of the ASMA at 370, 390 and 410 K. The red line denotes the slope of the original time-series. Two-sided p-value is showed in the upper-right corner of each subfigure.

---

## Author Response (AR2)

**Author Comment to Referee #1**

Egusphere-2025-1670, 'Interannual variability of the Asian summer monsoon anticyclone' by O. Kachula et al.

We thank Referee #1 for the positive review and valuable comments for the revised manuscript. Our reply to the reviewer comments is listed in detail below. Questions and comments of the referee are shown in bold face.

1. Q: I want to clarify the question, since I did not phrase it clearly. You use the optimisation method to find the background value which minimises the negative of the quotient of the Montgomery stream function (MSF) inside the ASMA box [0° – 90° N,0° – 180° E] vs outside the ASMA box [rest] +1. And you describe two cases, one where many MSF values inside the ASMA box are larger than outside (Fig. 1 of the manuscript), and one where there are MSF values outside of the ASMA box which are of comparable size to those inside (Fig. 2 of the manuscript). I would assume case one occurs most of the time during the peak phase of the ASMA in JJA, while the second case happens at the ASMA's start and end. I wondered whether the optimised background value during the peak phase of the ASMA is typically comparable to the maximum MSF value outside the ASMA region.

A: It is indeed possible to find a day, especially during the ASMA peak phase, when the background value would be equal to the maximum value outside of the ASMA box. But there is no guarantee that such favourable condition would cover all data. This leads to a set of heuristic rules when different treatment is required or a unified approach as the one proposed in this work.

2. Q: And secondly, I would like you to elaborate and to clarify why your method provides physically robust results to define the ASMA region, and why your results should be more accurate than those of Plöger et al. 2015, Santee et al. 2017 or Manney et al. 2021. I think this is important, especially because your ASMA area trend is negative in contrast to previous studies. What troubles me is that your optimisation method to derive the boundaries of the ASMA region is not based on physical quantities or gradients in the area, but on a comparison of MSF on the globe. With which physical reasoning should the strength of MSF values, e.g. over North America, affect the boundaries of the ASMA?

A: This is a purely mathematical method. The evidence of its applicability to solve the problem of finding the boundary of ASMA can be gathered through statistical analysis and comparison with the existing methodologies. The moment quantities provide the information about the spatio-temporal evolution of the ASMA. The explanation of why the method provides improvements to Santee et al. 2017 and Manney et al. 2021 was given in the manuscript with Figure 3 (also answered with Q2 in the response to Referee #1). Figure C2 of the manuscript highlights situations when the proposed method determines the boundary in contrast to Plöger et al. 2015 that provide the interpolated PV barrier value for this day.

3. Q: I saw that you removed (line 120 in the track changes) text pointing to the meaning of the ASMA boundary. Is there a reason for that?

A: Because in the manuscript we do not work with trace gases and aerosols hence the claim was removed.

4. Q: Some clarification would help me here. My question is whether the optimised background value changes when area weighting is applied in the calculation. This is not answered by Figure 4 in your reply, by area weighting the MSF on a map plot, and cannot be addressed by just stating the choice of a different projection. However, your reply ("At the same time after applying area weighting, we see a difference...") suggests that you also performed some tests, including area weighting in calculating the optimised background value. If this was done, a discussion of the sensitivity of the results to this adaptation should be added to the manuscript if the unweighted calculations are further used, or the entire calculations may be done area-weighted for the whole manuscript. In my opinion, some area weighting should be included in the calculation of the absolute vortex moment and in optimising the background value. In the updated manuscript, the area is weighted by applying the Lambert Cylindrical Equal-Area projection "in our regional calculations". What are the regional calculations meant by and affected by it? And any influence by the projection is not represented in the equations that should be affected. Matthewman et al. 2009 also introduced a projection to account for area weighting. There it seems that the projection is included in the dx dy (Eq. 5), for the absolute vortex momentum. It may also be included in your equation indirectly, but for me, it is harder to see (e.g. with delta x and delta y being 1). If that is the case, I would suggest clarifying your calculation.

A: We are sorry for the confusion to this matter. "in our regional calculations" means that we applied area weighting to find the optimised background value based on the regions ratio. Delta x and delta y being 1 are just stated to highlight that we work with the grid of size  $181 \times 360$ .

Figures 1 and 2 of this reply show the difference between optimised background value calculated with and without area weighting. As you can see the difference is biased toward positive values which tells us that the area weighting affects the optimized background values by slightly increasing them. But for majority of the background values the change is negligible. It also can be noticed that the higher standard deviation is caused by contribution from the outside of the ASMA active phase.

We also agree that the area weighting is needed for the moments quantities (centroid position, excess kurtosis, etc.) determination. We updated the plots in the manuscript. The analysis remains the same, the trend lines were not affected by the area weighting procedure. The moment quantities have slightly changed. The biggest change can be seen for the centroid longitude and excess kurtosis. The area of the ASMA was not affected at all because it was calculated using the polygon of the ASMA boundary instead of following Matthewman et al. 2009 approach (it is mentioned in the manuscript, see L239). To clarify this issue we updated the manuscript.

The rest of the comments were taken into account and corrected accordingly.

Figure 1: The yearly mean difference of the original  $\mu_b$  and the one calculated without area weighting  $(\mu_b^w)$  at 390 K.

Figure 2: The daily mean difference of the original  $\mu_b$  and the one calculated without area weighting  $(\mu_b^w)$  at 390 K.